… nature communications

# Security of quantum key distribution from generalised entropy accumulation

Tony Metger [1] ✉ & Renato Renner [1]

The goal of quantum key distribution (QKD) is to establish a secure key between two parties connected by an insecure quantum channel. To use a QKD protocol in practice, one has to prove that a finite size key is secure against general attacks: no matter the adversary's attack, they cannot gain useful information about the key. A much simpler task is to prove security against collective attacks, where the adversary is assumed to behave identically and independently in each round. In this work, we provide a formal framework for general QKD protocols and show that for any protocol that can be expressed in this framework, security against general attacks reduces to security against collective attacks, which in turn reduces to a numerical computation. Our proof relies on a recently developed information-theoretic tool called generalised entropy accumulation and can handle generic prepare-and-measure protocols directly without switching to an entanglement-based version.

Quantum key distribution (QKD) considers the following scenario: two parties, Alice and Bob, can communicate via an insecure quantum channel and an authenticated classical channel. An insecure quantum channel allows the adversary to intercept and tamper with any quantum state sent across the channel; an authenticated classical channel is one where an adversary can read every message sent across the channel, but cannot impersonate either party; for example, the adversary cannot convince Bob that a certain message was sent by Alice when in fact, it was not. Using these resources, Alice and Bob would like to establish a secure shared key, i.e. a piece of information that is known to both of them, but entirely unknown to an adversary Eve[1,2].

The key difficulty in establishing the security of a QKD protocol is that one has to take into consideration any possible attack that the adversary Eve may perform. For example, in one round of the protocol, Eve may gather a piece of quantum side information about the quantum state sent via the insecure channel. This piece of side information could be combined with side information from previous rounds to plan Eve's attack for the next round, resulting in a very complicated multi-round attack. Additionally, Alice and Bob can only execute a certain finite number of rounds, introducing statistical finite-size effects. A security proof that takes both of these challenges into account is called a finite-size security proof against general attacks (also referred to as coherent attacks)[3–5]. Such a proof is required to safely deploy a QKD protocol in practice.

Due to the difficulty of proving finite-size security against general attacks, many protocols are first analysed for collective attacks, for which very general numerical techniques have been developed (see e.g.[6–14]). For a security proof against collective attacks one makes the assumption that Alice and Bob execute infinitely many rounds of the protocol and Eve behaves independently and identically in each round. This is also called the i.i.d. asymptotic setting. These assumptions are of course unrealistic, but a collective attack proof is a useful theoretical tool as it can often be converted into a finite-size proof against general attacks; this is called a reduction to i.i.d.

There are a number of existing techniques for performing such a reduction to i.i.d. These techniques are very powerful, but typically require additional assumptions on the protocol and can significantly lower the amount of key that can be extracted compared to the collective attack scenario. The most widely used ones are either based on the quantum de Finetti theorem[15] (and the related post-selection technique[16]) or the entropy accumulation theorem (EAT)[17].

The quantum de Finetti theorem and related methods such as the post-selection technique rely on the permutation-symmetry between different rounds of the protocol to reduce general to collective attacks. While not every protocol possesses this permutation symmetry naturally, it can usually be enforced by including an additional "symmetrisation step" in the protocol. The main downside of these techniques is that the bounds they achieve scale unfavourably with the

[1]Institute for Theoretical Physics, ETH Zürich, 8093 Zürich, Switzerland. ✉e-mail: tmetger@ethz.ch

dimension of the underlying Hilbert space, i.e. the Hilbert space that contains the states sent from Alice to Bob. This means that these techniques only yield useful bounds for protocols with a small Hilbert space dimension, e.g. the BB84 or B92 protocols[1,18]. However, practical implementations of QKD protocols do not always satisfy this requirement; for example, many protocols use laser pulses as the means by which Alice sends a quantum state to Bob[19,20], and such laser pulses are described in a Fock space whose dimension is in principle unbounded. While methods for truncating the Fock space have been developed[21], this introduces additional complications and may lead to weak bounds if the dimension of the truncated Fock space remains large.

In contrast, the EAT provides bounds that do not depend on the dimension of the underlying Hilbert space. This dimension-independence of the second-order terms means that the EAT can also be used to prove security for device-independent or semi-device-independent protocols[22].

The main downside of the EAT for security proofs is that it requires that new side information must be output in a round-by-round manner subject to a Markov condition between rounds, and once side information has been output it cannot be updated anymore. In general, it is not possible to model the way that Eve actively intercepts quantum states and updates her side information in a prepare-and-measure protocol by a process that outputs side information in a round-by-round manner subject to the Markov condition. As a consequence, the EAT cannot "naturally" deal with general prepare-and-measure protocols. Instead, one first has to convert a prepare-and-measure protocol into an entanglement-based protocol. This can be done as follows: if Alice prepares one of a set of pure states $\left\{\left|\psi^j\right\rangle_Q\right\}_j$ with probability $p(j)$ and stores the index $j$ specifying the state in her register $A$, we can replace this by Alice preparing a state $\left|\tilde{\psi}\right\rangle_{AQ} = \sum_j \sqrt{p(j)}\left|j\right\rangle_A\left|\psi^j\right\rangle_Q$ and later measuring her system $A$. Then, we can model Eve's attack by replacing this state $\left|\tilde{\psi}\right\rangle_{AQ}$ by an arbitrary state $\left|\hat{\psi}\right\rangle_{AQE}$ prepared by Eve, subject to the constraint that Alice's marginal, which Eve cannot access in the prepare-and-measure protocol, is "correct", i.e. $\tilde{\psi}_A = \hat{\psi}_A$. This additional constraint is an artificial one in the sense that it is not something that Alice and Bob check in the actual protocol, and it is unclear how it can be incorporated into a security proof using the EAT in a natural way. As a result, it appears difficult or impossible to use the EAT to obtain reasonable finite-size key rates for prepare-and-measure protocols except in very simple cases.

In addition to these general techniques for reducing security against general attacks to security against collective attacks, there are also more specialised techniques that directly prove security against general attacks without an explicit reduction to collective attacks. Perhaps the most common of these are phase-error correction and entropic uncertainty techniques, both of which use the complementarity of different measurements in the protocol as the starting point for a security proof (see e.g., refs. [23–28]). These security proofs usually give very tight bounds for "symmetric" protocols (i.e. protocols relying on mutually unbiased measurement bases, even though these bases need not be chosen with equal probability) where they can be applied naturally, and can also be extended to symmetric protocols with experimental imperfections that slightly break the symmetry, e.g. using the reference state technique[29,30]. In addition, various other proof techniques that use the symmetry of specific protocols have been developed (see e.g. refs. [31–33]).

In this work, we show that security against collective attacks implies finite-size security against general attacks for a broad class of protocols. The main feature of our security proof is its generality: while many existing security proofs work well for particular protocols, our approach works for any generic protocol satisfying a few structural assumptions. Furthermore, it provides a natural way of proving security against general attacks, with the proof being in close correspondence to the structure of the original protocol, whereas previous techniques often required the protocol to be transformed into a theoretically equivalent one to fit into the framework of a particular proof technique. In particular, our technique can be applied directly to prepare-and-measure protocols without transforming them into an entanglement-based version. As a sample application, we show that a direct application of our general framework yields the first asymptotically tight finite-size security proof against general attacks for the B92 protocol. Importantly, our technique provides bounds that are independent of the dimension of the underlying Hilbert space; instead, the bound depends only on the number of possible classical outputs that Alice and Bob may receive. This is particularly relevant for photonic QKD protocols, where the underlying Hilbert space is a Fock space with unbounded dimension[34,35], and is also useful for (semi-)device-independent protocols. For our security proof, we employ the generalised entropy accumulation theorem (GEAT), a recent information-theoretic result[36] that resembles the EAT discussed above, but allows a more flexible model of side information; this enables us to circumvent many of the difficulties in applying the EAT and deal with prepare-and-measure protocols directly, while retaining the advantages of the EAT, most importantly its dimension-independence.

## Results

### Framework for prepare-and-measure protocols

Our main result, Theorem 4, shows that for a broad class of prepare-and-measure protocols, security against collective attacks implies security against general attacks. To make this result easy to use, we phrase it as a security statement for a general "template protocol"; many existing prepare-and-measure protocols can be viewed as an instance of this template protocol, and their security then follows from the security of the general template protocol. For protocols that do not fit exactly into this template, the security proof can usually easily be adapted from our proof of Theorem 4.

Our template protocol is described formally in Box 1; here, we make a few additional remarks regarding this general protocol, using the notation introduced in Box 1. Firstly, without loss of generality, we can assume that the cq-state $\psi_{UQ}$ is of the form $\psi_{UQ} = \sum_u p(u)|u\rangle\langle u| \otimes |\psi\rangle\langle\psi|_{Q|u}$ for a probability distribution $p(u)$ and pure states $|\psi\rangle\langle\psi|_{Q|u}$. This means that Alice chooses a value $u$ according to $p(u)$ and then sends the pure state $|\psi\rangle\langle\psi|_{Q|u}$ to Bob. The reason that we can assume that $|\psi\rangle\langle\psi|_{Q|u}$ is pure is that if Alice wanted to send a mixed state, she could express that mixed state as a mixture of pure states, send one of those pure states, and later "forget" which of the pure states she sent as part of the map RK.

Secondly, in the protocol in Box 1, Bob measures a POVM $\{N^{(v)}\}$ with outcomes $v \in \mathcal{V}$. More commonly, we think of Bob as choosing an input $y$ according to some distribution $q(y)$ and receiving an output $b \in \mathcal{B}$. This can be described by a collection of POVMs $\{\tilde{N}_y^{(b)}\}_{b \in \mathcal{B}}$, one for each possible input $y$. For example, Bob might choose uniformly at random whether to measure a qubit in the computational or Hadamard basis. In that case, $y$ would be the basis choice, and for each $y$, $\{\tilde{N}_y^{(b)}\}_{b \in \mathcal{B}}$ is the measurement in the chosen basis. However, since Bob's measurements are trusted, the distinction between inputs and outputs is unnecessary: we can convert a set of POVMs $\{\tilde{N}_y^{(b)}\}_{b \in \mathcal{B}}$ with an input distribution $q(y)$ into an equivalent single POVM $\{N^{(v)}\}_{v \in \mathcal{V}}$ by choosing $\mathcal{V} = \mathcal{Y} \times \mathcal{B}$ and $N^{(y,b)} = q(y)\tilde{N}_y^{(b)}$. This satisfies the required property of a POVM:

$$\sum_{y,b} N^{(y,b)} = \sum_y q(y) \sum_b \tilde{N}_y^{(b)} = \sum_y q(y)\mathbb{1} = \mathbb{1}, \tag{1}$$

## BOX 1

# General prepare-and-measure QKD protocol

**Protocol arguments:**

$\mathbf{n} \in \mathbb{N}$ : number of rounds.

$\psi_{UQ}$ : quantum state prepared by Alice, where $U$ is classical with alphabet $\mathcal{U}$ and $Q$ is quantum.

$\{\mathbf{N^{(v)}}\}_{\mathbf{v} \in \mathcal{V}}$ : POVM acting on Hilbert space $\mathcal{H}_{\mathbf{Q}}$ describing Bob's trusted measurements (where $\mathcal{V}$ is some finite set of possible outcomes).

PD : $\mathcal{U} \times \mathcal{V} \to \mathcal{I}$ : function describing transcript of public discussion (where $\mathcal{I}$ is some finite alphabet).

RK : $\mathcal{U} \times \mathcal{I} \to \mathcal{S}$ : function describing Alice's raw key generation (where $\mathcal{S}$ is the alphabet of the raw key).

EV : $\mathcal{V} \times \mathcal{I} \times \mathcal{S} \to \mathcal{C}$ : function "evaluating" each round by assigning a label from the alphabet $\mathcal{C}$

$\lambda_{EC} \in \mathbb{N}_0$ : length of bit string exchanged during error correction step.

$k_{CA} > 0$ : required amount of single-round entropy generation.

$\varepsilon_{KV}, \varepsilon_{PA} > 0$ : tolerated errors during key validation and privacy amplification steps.

CA : $\mathbb{P}(\mathcal{C}) \to \mathbb{R}$ : affine function corresponding to collective attack bound.

$\mathbf{l} \in \mathbb{N}$ : length of final key.

**Protocol steps:**

(1) Data generation: Alice prepares $\psi_{\mathbf{U^n Q^n}} = \psi_{\mathbf{UQ}}^{\otimes \mathbf{n}}$ and sequentially sends the systems $Q_1, ..., Q_n$ to Bob via a public quantum channel. For each $i \in \{1, ..., n\}$, Bob measures $\{\mathbf{N^{(v)}}\}_{\mathbf{v} \in \mathcal{V}}$ on register $Q_i$ and records the outcome in register $V_i$.

(2) Public discussion: for each $i \in \{1, ..., n\}$, Alice and Bob publicly exchange information $I_i = \mathrm{PD}(U_i, V_i)$.

(3) Raw key generation: for each $i \in \{1, ..., n\}$, Alice computes $S_i = \mathrm{RK}(U_i, I_i)$.

(4) Error correction: Alice and Bob publicly exchange information EC $\in \{0,1\}^{\lambda_{EC}}$, which can depend on $U^n$, $V^n$, and $I^n$. Bob computes $\hat{\mathbf{S}}^{\mathbf{n}}(\mathrm{EC}, \mathbf{V^n}, \mathbf{I^n}) \in \mathcal{S}^{\mathbf{n}}$.

(5) Raw key validation: Alice chooses a function HASH : $\mathcal{S}^{\mathbf{n}} \to \{0,1\}^{\lceil \log(1/\varepsilon_{KV}) \rceil}$ from a universal hash family $\mathcal{F}$ (Definition 5) according to the associated probability distribution $\mathbf{P}_{\mathcal{F}}$ and publishes a description of HASH and the value HASH$(S^n)$. Bob computes HASH$(\hat{\mathbf{S}}^{\mathbf{n}})$ and aborts the protocol if HASH$(\mathbf{S^n}) \neq$ HASH$(\hat{\mathbf{S}}^{\mathbf{n}})$.

(6) Statistical check: for each $i \in \{1, ..., n\}$, Bob sets $\hat{\mathbf{C}}_{\mathbf{i}} = \mathrm{EV}(\mathbf{V_i}, \mathbf{I_i}, \hat{\mathbf{S}}_{\mathbf{i}})$. Bob then computes $\mathbf{k} = \mathrm{CA}(\mathrm{freq}(\hat{\mathbf{C}}^{\mathbf{n}}))$. If $k < k_{CA}$, he aborts the protocol.

(7) Privacy amplification: Alice and Bob convert their registers $S^n$ and $\hat{\mathbf{S}}^{\mathbf{n}}$ to a binary representation, obtaining strings of length $m$. Alice chooses a seed $\mu \in \{0, 1\}^m$ uniformly at random and publishes her choice. Alice and Bob compute $l$-bit strings $K = \mathrm{EXT}(S^n, \mu)$ and $\hat{\mathbf{K}} = \mathrm{EXT}(\hat{\mathbf{S}}^{\mathbf{n}}, \mu)$, respectively, where EXT: $\{0, 1\}^m \times \{0, 1\}^m \to \{0, 1\}^l$ is a quantum-proof strong $(\mathbf{l} + \lceil 2 \log(1/\varepsilon_{PA}) \rceil, \varepsilon_{PA})$-extractor (Definition 6).

where we used the fact that $\{\bar{N}_y^{(b)}\}_{b \in \mathcal{B}}$ is a POVM for the first equality and the fact that $q(y)$ is a probability distribution for the second. One can think of $N^{(y,b)}$ as first choosing $y \in \mathcal{Y}$ according to $q(y)$ and then measuring $\{\tilde{N}_y^{(b)}\}$ on the state, providing $(y, b)$ as output.

Thirdly, the function PD describes the total information exchanged during the public discussion (Step (2)) for one round $i$ of the protocol. The details of how the public discussion takes place are of no concern to the protocol: in general, Alice and Bob may exchange multiple rounds of back-and-forth communication during this step, and PD describes the transcript of the entire exchange. For example, in a protocol that includes a sifting step, the public discussion would include the information necessary to decide which rounds to sift out; the actual sifting would occur in the raw key generation step, where Alice's function RK can use the information from the public discussion to put a special symbol (e.g. $\perp$) as the raw key for rounds that are sifted out.

Additionally, the protocol distinguishes between information $I_i$ published during Step (2) and error correction information EC published during Step (4). The difference between these two steps is that $I_i$ may only depend on the inputs $U_i$ and $V_i$ generated during the $i$-th round of measurements. This means that $I_i$ is generated in a round-by-round manner and will enter in the single-round security statement (or collective attack bound, see "Definition 2"). In contrast, EC is global information of a fixed length $\lambda_{EC}$, i.e. it can depend arbitrarily on information generated during all rounds of the protocol, but to obtain a good key rate, $\lambda_{EC}$ should be as short as possible. We note that the bound on the length of EC is needed in Supplementary Eq. (5), where we use it to remove the error correction information from the conditioning system; one can replace Supplementary Eq. (5) by a slightly more sophisticated chain rule that subtracts a (one-shot) mutual information between EC and $S^n$. In that case, the protocol needs to specify an upper bound on this mutual information instead of the length $\lambda_{EC}$.

Finally, we note that in the protocol in Box 1, Alice and Bob first perform error correction, and afterwards Bob uses his error-corrected guess for Alice's raw key for the purposes of the statistical check. An alternative that is commonly used in existing QKD protocols is that Alice and Bob publish part of their data in a separate parameter estimation step before the error correction step and use this public information to run a statistical check. Our protocol in Box 1 can easily be modified to include protocols of this form. For the modified protocol, the security proof stays exactly the same, except that the reduction from Theorem 4 to Claim 10 now follows almost trivially and does not need the argument from Supplementary Note A.

Example: BB84 protocol as an instance of Box 1. To gain further intuition for the protocol in Box 1, we describe how to reproduce the well-known BB84 protocol as an instance of our general protocol in Box 1. In the BB84 protocol, Alice sends a random state from the set $\{|0\rangle, |1\rangle, |+\rangle, |-\rangle\}$, where $|\pm\rangle = \frac{|0\rangle \pm |1\rangle}{\sqrt{2}}$ are the Hadamard basis states. As her information $U_i$, Alice records which state she sent, i.e. she records the basis $x \in \{0, 1\}$ and the value $a \in \{0, 1\}$. Hence, for the BB84 protocol,

$$\psi_{UQ} = \frac{1}{4} \sum_{x,a \in \{0,1\}} |x,a\rangle\langle x,a|_U \otimes H^x |a\rangle\langle a|_Q H^x, \tag{2}$$

where $H$ is the Hadamard gate and $H^0 = \mathrm{id}$, $H^1 = H$. Bob's measurements output a basis choice $y \in \{0, 1\}$ and the outcome $b$ of a single-qubit measurement in that basis (with $y = 0$ corresponding to the computational and $y = 1$ to the Hadamard basis). Therefore, his measurements are described by a POVM on system $Q$ consisting of elements

$$N^{(y,b)} = \frac{1}{2} H^y |b\rangle\langle b| H^y. \tag{3}$$

During the public discussion phase, Alice and Bob publish their basis choices $x_i$ and $y_i$ for each of the rounds. Therefore, for $U_i = (x_i, a_i)$ and $V_i = (y_i, b_i)$,

$$I_i = \text{PD}(U_i, V_i) = (x_i, y_i). \qquad (4)$$

To generate her raw key, for each round Alice checks whether the basis choices $x_i$ and $y_i$ are the same: if so, she uses her measurement outcome $a_i$ for the raw key, and otherwise she discards that round. Formally,

$$S_i = \text{RK}(U_i, I_i) = \text{RK}((x_i, a_i), (x_i, y_i)) = \begin{cases} a_i & \text{if } x_i = y_i, \\ \perp & \text{otherwise.} \end{cases} \qquad (5)$$

Finally, for the statistical check in Step (6), Bob checks whether his guess $\hat{S}^n$ for Alice's string matches his own raw data. In fact, Bob can only do this check on a small subset of indices $i$. The reason is that for our definition of collective attack bounds ("Definition 2") and the security proof (Theorem 4), we are bounding the entropy conditioned on the systems $C^n$, i.e. we are essentially assuming that all of the statistical information gets leaked to Eve. Hence, Bob chooses a value $T_i$ at random with $\Pr[T_i = 1] = \gamma$ (where $\gamma$ is the testing probability, and the choice of $T_i$ can formally be included into $V_i$), and then sets

$$\hat{C}_i = \text{EV}(V_i, I_i, \hat{S}_i) = \text{EV}((y_i, b_i), (x_i, y_i), \hat{S}_i) \qquad (6)$$

$$= \begin{cases} \perp & \text{if } x_i \neq y_i \text{ or } T_i = 0, \\ 1 & \text{if } x_i = y_i, T_i = 1, \text{and } b_i = \hat{S}_i, \\ 0 & \text{otherwise.} \end{cases} \qquad (7)$$

Intuitively, $\perp$ denotes that no useful check can be performed in this round, "1" means the check has passed, and "0" means the check has failed.

## Modelling Eve's attack

In the protocol in Box 1, Eve can obtain information about the final key $K$ in two ways: firstly, Eve can observe the classical information published by Alice and Bob during the protocol, e.g. the error correction information EC. In a security proof, this is easy to handle, as Alice and Bob have full control over what information they publish. Secondly, Eve can intercept the quantum systems $Q_i$ sent from Alice to Bob in Step (1). This is much harder to analyse in a security proof as Eve can perform arbitrary operations on the systems $Q_i$ and we need to bound the amount of information Eve can gain about Alice's and Bob's raw key from tampering with the systems $Q_i$ without being detected. The set of actions Eve performs on the systems $Q_i$ is called Eve's attack.

In principle, Eve could collect all of the $n$ systems $Q_1, …, Q_n$, perform an arbitrary quantum channel $\mathcal{A} : Q^n \rightarrow EQ^n$, and send the output on systems $Q^n$ to Bob. The system $E$ would be kept by Eve and would contain her (potentially quantum) side information about the final key.

To analyse the security of a prepare-and-measure protocol with the GEAT, we need to introduce an extra condition.

**Condition 1.** Eve can only be in possession of one of the systems $Q_i$ at the same time.

Since Alice sends the systems $Q_1, …, Q_n$ sequentially in Step (1), this means that with this additional condition, Eve's most general attack also takes a sequential form. More formally, with this condition, the most general attack Eve can perform is described by a sequence of maps $\mathcal{A}_i : E'_{i-1} Q_i \rightarrow E'_i Q_i$, where $E'_i$ are arbitrary quantum systems that contain Eve's side information after having intercepted the $i$-th system $Q_i$. (The system $E_0$ can be chosen to be trivial without loss of generality, but we will not need this for our security proof).

In fact, it is easy for Alice and Bob to enforce Condition 1 by checking that system $Q_i$ has arrived on Bob's side before $Q_{i+1}$ is sent. The downside of this simple strategy is that if Alice and Bob are far apart, it limits the number of signals that can be sent per unit time.

To circumvent this, Alice and Bob can agree on a "schedule" on which signals are transmitted, i.e. they decide when Alice will send out each signal, so Bob, being aware of its travel time without Eve's interference, knows when to expect to receive it. Then, assuming that Eve cannot significantly speed up the transmission of signals, this would ensure that Condition 1 is satisfied without Alice having to wait for Bob's confirmation to send the next signal (see Supplementary Fig. 1 for an illustration of this). Whether or not the assumption that Eve cannot significantly speed up the transmission of signals is realistic depends on the specific QKD setup: for example, if signals are transmitted from Alice to Bob through vacuum (e.g. in satellite-to-satellite QKD), they travel at the speed of light and cannot be sped up further by Eve, so Condition 1 can be enforced by sending signals on a pre-agreed schedule without issues.

On the other hand, if Alice and Bob exchange signals via a (very long) optical fibre, Eve could in principle extract the signal at the start of the fibre, transmit it through free space, and then re-insert it into the fibre on Bob's side. Since the speed of light in a fibre is slower than in free space, this would enable Eve to have simultaneous access to a (relatively small) set of $s$ sped-up signals, perform some attack involving this set of signals, and then feed the "first" of these signals to Bob in such a way that it arrives at the time expected by Bob; then, Eve could add the next sped-up signal to her set, apply another attack to that set of $s$ signals, and so on. Such an attack would violate Condition 1, but it would go unnoticed by Alice and Bob since the signals do arrive at the expected times on Bob's end.

Setting aside the question of how realistic it is for Eve to perform such an attack, this issue can be addressed by relaxing Condition 1 so that instead of requiring Eve to be in possession of only one signal at a time, we allow her to be in possession of $s$ signals at a time. To prove security under this weakened condition, we can divide the signals into interleaved groups such that any two signals within a group are $s$ rounds apart, use a standard chain rule for min-entropies (or Renyi entropies) to divide the total entropy into a sum of group-wise entropies, and simply apply our analysis at the level of these groups. Our proof then goes through essentially unchanged, although the resulting second-order terms in the key rate will depend on the allowed number $s$ of signals available to Eve at a time. We explain this modification in more detail in Supplementary Note C and focus on the case where Condition 1 holds exactly in the main text for simplicity.

We have now seen how to model Eve's general attack under Condition 1. In contrast to such general sequential attacks, collective attacks only allow Eve to perform the same independent attack in each round of the protocol. Hence, a collective attack can be modelled by a map $\mathcal{A} : Q \rightarrow EQ$, which Eve applies in each round of the protocol, so Eve's full attack over $n$ rounds is given by the tensor product map $\mathcal{A}^{\otimes n} : Q^n \rightarrow E^n Q^n$. Proving security against this restricted class of attacks is typically much easier than proving security against general attacks. However, we stress that, unlike Condition 1, the assumption that Eve performs only a collective attack cannot be enforced by Alice and Bob. Therefore, a security proof that only considers collective attacks is insufficient for practical applications.

## Collective attack bounds

If one restricts Eve to performing collective attacks, it is known that in the limit $n \rightarrow \infty$ of many rounds the key rate is given by a simple entropic expression that only involves quantities corresponding to a single round of the protocol[37]. Note that the entropic expression for the key rate in[37] already includes information leaked to Eve during the error correction step assuming an optimal error correcting protocol. Our Definition 2 does not include a term corresponding to this –

instead in Box 1 we assume that the error correction information has length at most $\lambda_{EC}$, which we can later subtract from the length of the final key that can be generated.

More formally, we can view a collective attack bound as a map that takes as input the statistics corresponding to a single round of the protocol and outputs a lower bound on a certain conditional entropy, which specifies how much key can safely be extracted from a state with those statistics.

**Definition 2.** (Collective attack bound for Box 1) Fix arguments $\psi_{UQ}, \{N^{(v)}\}_{v \in \mathcal{V}}, PD, RK$, and EV for the protocol in Box 1. Suppose that Alice and Bob run a single round (i.e. $n = 1$) of the protocol in Box 1 with these arguments up to (and including) Step (3). For a collective attack $\mathcal{A} : Q \to QE$, denote the state at the end of Step (3) as $\nu_{UVSIE}$. Let $\nu_{UVSIEC}$ be an extension of this state, where $C = EV(V, I, S)$. A collective attack bound (for the choice of parameters fixed above) is a map $CA : \mathbb{P}(\mathcal{C}) \to \mathbb{R}$ such that for any collective attack $\mathcal{A}$, the state $\nu_{UVSIEC}$ (which depends on $\mathcal{A}$) satisfies

$$H(S|IEC)_\nu \geq CA(\nu_C). \tag{8}$$

## Security against general attacks

Having introduced our framework for general prepare-and-measure protocols and collective attack bounds, we can now state the main technical result of this paper, namely that a collective attack bound implies a security statement against general attacks. For this, we first recall the security definition for QKD, namely the notions of correctness, secrecy, and completeness[15]. This security definition is composable, meaning that the key generated by a protocol satisfying this definition can safely be used for other protocols[38].

**Definition 3.** (Correctness, secrecy, and completeness) Consider a QKD protocol in which Alice and Bob can decide whether or not to abort the protocol. Let $\rho_{K\hat{K}E}$ be the final state at the end of the protocol (for a given initial state), where $K$ and $\hat{K}$ are Alice's and Bob's version of the final key, respectively, and $E$ contains all side information available to the adversary Eve at the end of the protocol. The protocol is called $\varepsilon^{cor}$-correct, $\varepsilon^{sec}$-secret, and $\varepsilon^{comp}$-complete if the following holds:

(i)  Correctness. For any actions of the adversary Eve:

$$\Pr\left[K \neq \hat{K} \wedge \text{ not abort}\right] \leq \varepsilon^{cor}. \tag{9}$$

(ii)  Secrecy. For any actions of the adversary Eve:

$$\left\| \rho_{KE \wedge \Omega} - \tau_K \otimes \rho_{E \wedge \Omega} \right\|_1 \leq \varepsilon^{sec}, \tag{10}$$

where $\tau_K$ is the maximally mixed state on system $K$, $\Omega$ is the event that the protocol does not abort, and $\rho_{\wedge\Omega} = \Pr[\Omega]\rho_{|\Omega}$ is the subnormalised state conditioned on $\Omega$ (see Methods Subsection "Notation" for details). Note that here and throughout the paper, we use the difference in trace norm, not the trace distance. The latter has an additional normalisation factor of $\frac{1}{2}$.

(iii)  Completeness. For a given noise model for the protocol there exists an honest behaviour for the adversary Eve such that

$$\Pr\left[\text{abort}\right] \leq \varepsilon^{comp}. \tag{11}$$

Note that correctness and secrecy must hold for any behaviour of Eve (and also any noise model), while completeness is concerned with the honest implementation of the protocol. Correctness and secrecy bound the probability of Alice and Bob receiving different or insecure keys without detecting this fact and aborting the protocol. Completeness says that the protocol is robust against a given noise model in the sense that for this noise model, the probability of aborting the

protocol is small if Eve behaves honestly. It is common to combine the correctness and secrecy parameters and call a protocol $(\varepsilon^{cor} + \varepsilon^{sec}/2)$-secure, where the factor of 1/2 arises because our definition of secrecy uses the difference in trace norm, not the trace distance, which has an additional factor of 1/2.

Our main result is that the protocol in Box 1 satisfies the correctness and secrecy conditions. Formally, we show the following.

**Theorem 4.** Fix any choice of arguments $n, \psi_{UQ}, \{N^{(v)}\}_{v \in \mathcal{V}}, PD, RK, EV, k_{CA}, \lambda_{EC}, \varepsilon_{KV}$, and $\varepsilon_{PA}$ for Box 1. Let $CA : \mathbb{P}(\mathcal{C}) \to \mathbb{R}$ be an affine collective attack bound for this choice of arguments. For any $\varepsilon_s, \varepsilon_a > 0$ and $\alpha \in (1, 3/2)$, choose a final key length $l$ that satisfies

$$l \leq n\, k_{CA} - n\, \frac{\alpha - 1}{2 - \alpha} \frac{\ln(2)}{2} V^2 - \frac{g(\varepsilon_s) + \alpha \log(1/\varepsilon_a)}{\alpha - 1}$$
$$- n \left( \frac{\alpha - 1}{2 - \alpha} \right)^2 K'(\alpha) - \lceil 2\log(1/\varepsilon_{PA}) \rceil - \lceil \log(1/\varepsilon_{KV}) \rceil - \lambda_{EC},$$
$$\tag{12}$$

where $g(\varepsilon_s), V$, and $K'(\alpha)$ are defined in Theorem 9. With this choice of parameters and assuming that Condition 1 holds, the protocol in Box 1 is $\varepsilon^{cor}$-correct and $\varepsilon^{sec}$-secret for

$$\varepsilon^{cor} = \varepsilon_{KV}, \qquad \varepsilon^{sec} = \max\{\varepsilon_{PA} + 4\,\varepsilon_s, 2\,\varepsilon_a\} + 2\,\varepsilon_{KV}. \tag{13}$$

We prove this theorem in "Methods" subsection "Proof of main theorem". In addition, we also show completeness; since this is much more straightforward and only uses standard techniques, we defer this to Supplementary Note B.

## Sample application: B92 protocol

We now demonstrate how to apply our framework, using the B92 protocol as an example. The B92 protocol has no natural entanglement-based analogue (i.e. an equivalent entanglement-based protocol that does not require "artificial" constraints on the reduced state on Alice's side and still achieves the same key rate as the prepare-and-measure version of B92) and therefore cannot be analysed with the original EAT. Nonetheless, the B92 protocol is very simple, and therefore provides arguably the easiest example to demonstrate the application of our framework to a protocol that cannot be analysed with the EAT. Furthermore, while there exist analytic security proofs of B92 using entropic uncertainty relations[25,39], these techniques yield key rates that are far from optimal even in the asymptotic regime. This is in contrast to highly symmetric protocols such as BB84, where entropic uncertainty relations yield essentially tight proofs[28].

We emphasise that the purpose of this section is to illustrate our general results with a simple example, not to derive the tightest possible key rates for a particular protocols. We leave the analysis of more complicated protocols, where deriving the collective attack bound may be more involved, for future work. In Supplementary Note G, we also sketch how to express the decoy state BB84 protocol as an instance of our framework and how to derive a collective bound for it, demonstrating that the widely-used decoy state technique also naturally fits within our framework.

We also note that very recent work[40] has analysed the performance of the EAT on entanglement-based QKD protocols (and prepare-and-measure protocols that have a natural entanglement-based analogue) and found that it provides better key rates than previous methods. Since our GEAT-based security proof produces essentially the same key rates as the EAT in cases where both methods can be applied, this suggests that our framework will provide very good key rates also in cases where the EAT cannot be applied.

We start by giving an informal description of the B92 protocol and the intuition behind it. Then, we show how to view the B92 protocol as

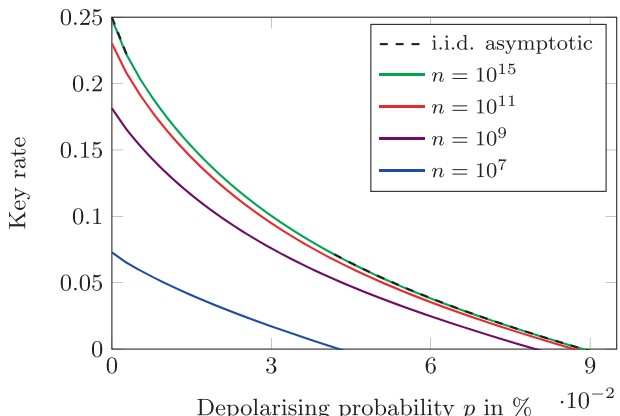

**Fig. 1 | Key rates for the B92 protocol as a function of the depolarising probability $p$ for $\varepsilon^{\mathrm{cor}} = 5 \cdot 10^{-11}, \varepsilon^{\mathrm{sec}} = 10^{-9}$, and $\varepsilon^{\mathrm{comp}} = 10^{-2}$.** The dashed line shows the key rate in the i.i.d. asymptotic setting, i.e. assuming that Eve behaves the same in each round and infinitely many rounds are executed. We see that as the number $n$ of rounds in the protocol increases, the finite-size key rates against general attacks approach the i.i.d. asymptotic rate.

an instance of our general protocol in Box 1. Using the technique from "Results" subsection "Collective attack bounds" to derive a collective attack bound, we can then apply Theorem 4 to obtain a security statement for general attacks. To illustrate the result, we numerically compute the key rate for different choices of the number of rounds and tolerated noise level in Fig. 1.

Each round of the B92 protocol works as follows: Alice chooses a bit $u \in \{0, 1\}$ uniformly at random. If $u = 0$, she prepares the state $|\psi\rangle_Q = |0\rangle$, whereas if $u = 1$, she prepares $|\psi\rangle_Q = |+\rangle$. She sends $|\psi\rangle_Q$ to Bob, who chooses $y \in \{0, 1\}$ uniformly at random and measures the system $Q$ in the computational basis if $y = 0$ and the Hadamard basis if $y = 1$. If he obtains outcome "1" (when measuring in the computational basis) or "-" (when measuring in the Hadamard basis), he sets $v = y \oplus 1$. Otherwise, he sets $v = \perp$. In the sifting step, Bob announces in which rounds he recorded $v = \perp$, and Alice sets $u = \perp$ for those rounds, too. The bits $u$ and $v$ from all of the rounds form the raw key. To detect possible tampering by Eve, Alice and Bob compare their values of $u$ and $v$ on a subset of rounds.

The intuition behind this protocol is the following: the secret information that will make up the key is encoded in Alice's basis choice $u$ (where $u = 0$ corresponds to the computational and $u = 1$ to the Hadamard basis). When Bob receives the system $Q$ he tries to find out which basis the state was prepared in. For this, he guesses a basis $y$ and measures $Q$ in this basis. Suppose he chose $y = 0$, i.e. the computational basis, and assume that Eve did not tamper with the system $Q$. Then, if he obtains outcome "1" he concludes that Alice cannot have prepared the state $|0\rangle$ and therefore must have chosen $u = 1$. Accordingly, he sets $v = 1 = y \oplus 1$. If Bob obtains outcome "0" he cannot deduce Alice's basis choice as both the states $|0\rangle$ and $|+\rangle$ may produce outcome "0" when measured in the computational basis, so he sets $v = \perp$. Likewise, if he chose $y = 1$ and obtains outcome "-", this provides conclusive evidence that Alice cannot have prepared the state $|+\rangle$, so he sets $v = 0 = y \oplus 1$, whereas the outcome "+" is inconclusive. If Eve tries to tamper with the system $Q$, she is likely to disturb the state as she does not know which basis it was prepared in. Therefore, Alice and Bob will detect this tampering when comparing their values of $u$ and $v$.

We now give a more formal description of the B92 protocol as an instance of the protocol in Box 1. As for the BB84 protocol described in Results Subsection "Framework for prepare-and-measure protocols", this means specifying the arguments $\psi_{UQ}, \{N^{(v)}\}_{v \in \mathcal{V}}, \mathrm{PD}, \mathrm{RK}$, and EV. For each round Alice chooses a bit $U_i$ uniformly at random and prepares $|0\rangle$

or $|+\rangle$ based on her choice, so

$$\psi_{UQ} = \tfrac{1}{2}(|0\rangle\langle 0|_U \otimes |0\rangle\langle 0|_Q + |1\rangle\langle 1|_U \otimes |+\rangle\langle +|_Q). \tag{14}$$

Bob measures in either the computational or Hadamard basis and uses the outcome to determine $V_i \in \{0, 1, \perp\}$ as described before. This measurement is described by the following POVM:

$$N^{(0)} = \tfrac{1}{2}|-\rangle\langle -|, \; N^{(1)} = \tfrac{1}{2}|1\rangle\langle 1|, \; N^{(\perp)} = \tfrac{1}{2}(|0\rangle\langle 0| + |+\rangle\langle +|). \tag{15}$$

During the public discussion phase, Bob informs Alice which rounds were inconclusive, i.e. yielded outcome $\perp$. Therefore,

$$I_i = \mathrm{PD}(U_i, V_i) = \begin{cases} \perp & \text{if } V_i = \perp, \\ \top & \text{otherwise.} \end{cases} \tag{16}$$

To generate her raw key $S^n$, Alice uses her bits $U_i$ and discards the rounds for which Bob's measurement outcome was inconclusive, which she knows from the value of $I_i$:

$$S_i = \mathrm{RK}(U_i, I_i) = \begin{cases} \perp & \text{if } I_i = \perp, \\ U_i & \text{otherwise.} \end{cases} \tag{17}$$

To generate the statistics $\hat{C}_i$, Bob will check whether his guess $\hat{S}^n$ for Alice's raw key agrees with his own raw data $V^n$. As for the BB84 protocol described in Results Subsection "Framework for prepare-and-measure protocols", Bob can only do so on a small fraction $\gamma$ of rounds because Definition 2 includes the classical statistics as a conditioning system. Therefore, Bob chooses a value $T_i$ at random with $\Pr[T_i = 1] = \gamma$ (the choice of $T_i$ can formally be included into $V_i$ or one can view EV as a randomised rather than deterministic function). If $T_i = 0$, he sets $\hat{C}_i = \perp$, i.e. $\mathrm{EV}_{T_i=0}(V_i, I_i, \hat{S}_i) = \perp$. Otherwise, he sets $\hat{C}_i = \mathrm{EV}_{T_i=1}(V_i, I_i, \hat{S}_i)$ to

$$\begin{cases} \texttt{fail} & \text{if } \hat{S}_i = 0 \wedge V_i = 1 \text{ or } \hat{S}_i = 1 \wedge V_i = 0, \\ \texttt{inc} & \text{if } V_i = \perp, \\ \varnothing & \text{else.} \end{cases} \tag{18}$$

Of course, the functions $\mathrm{EV}_{T_i=0}$ and $\mathrm{EV}_{T_i=1}$ can be combined into a single function EV to formally fit into the framework of Box 1.

We need to derive an affine collective bound $\mathrm{CA}(\nu_C) = \vec{\lambda} \cdot \vec{\nu}_C + c_{\vec{\lambda}}$ for the B92 protocol, where $\vec{\nu}_C$ denotes the probability vector of distribution $\nu_C$ as in Results Subsection "Collective attack bounds". For this, we use the steps and notation from Methods Subsection "Deriving collective attack bounds"; we recommend skipping this subsection on a first reading and returning to it after understanding that subsection.

In the notation of Methods Subsection "Deriving collective attack bounds", the state $\tilde{\psi}_{PQ}$ is given by

$$\tilde{\psi}_{PQ} = \tfrac{1}{\sqrt{2}}(|0\rangle_P \otimes |0\rangle_Q + |1\rangle_P \otimes |+\rangle_Q). \tag{19}$$

For any state $\hat{\psi}_{PQ}$ chosen by Eve, the statistics observed by Alice and Bob are described by

$$\vec{\nu}_C = \mathrm{Tr}\left[\vec{\Gamma} \hat{\psi}_{PQ}\right], \tag{20}$$

where $\vec{\Gamma} = (\Gamma_{\texttt{fail}}, \Gamma_{\texttt{inc}}, \Gamma_{\varnothing}, \Gamma_{\perp})$ with

$$\Gamma_{\texttt{fail}} = \gamma(|0\rangle\langle 0|_P \otimes N_Q^{(1)} + |1\rangle\langle 1|_P \otimes N_Q^{(0)}), \tag{21}$$

$$\Gamma_{\texttt{inc}} = \gamma \mathbb{1}_P \otimes N_Q^{(\perp)}, \tag{22}$$

$$\Gamma_\varnothing = \gamma(\mathbb{1} - \Gamma_{\texttt{fail}} - \Gamma_{\texttt{inc}}), \tag{23}$$

$$\Gamma_\perp = (1 - \gamma)\mathbb{1}_P \otimes \mathbb{1}_Q, \tag{24}$$

and $\mathrm{Tr}\left[\vec{\Gamma}\,\hat{\psi}_{PQ}\right]$ is shorthand for the vector of the traces with the individual elements of $\vec{\Gamma}$. We can now directly apply the method from Methods Subsection "Deriving collective attack bounds" to find a collective attack bound $\mathrm{CA}(\nu_C) = \vec{\lambda} \cdot \vec{\nu}_C + c_{\vec{\lambda}}$: we can heuristically choose a $\vec{\lambda}$ and then determine $c_{\vec{\lambda}}$ by solving the convex optimisation problem from Equation (70) using the package Matlab CVXQUAD[41]. Note that one can pick $\vec{\lambda}$ by any numerical optimisation technique such as Matlab's fminsearch: since $\vec{\lambda}$ can be chosen heuristically, it is not an issue if such an optimisation method does not have a convergence guarantee. In contrast, to determine $c_{\vec{\lambda}}$ one must use an optimisation method that guarantees a lower bound in order to ensure that the collective attack bound is valid. This is why it is important that $c_{\vec{\lambda}}$ be determined via a convex optimisation problem for which one can certify the solution by duality. For our numerical implementation, we employ additional simplifications to the optimisation problem from Equation (70) using the steps described in Supplementary Note E. This helps with numerical performance, but is not strictly necessary.

As our noise model for an honest implementation, we consider the depolarising channel with depolarising probability $p$, i.e. the channel that maps $\rho \mapsto (1-p)\rho + p\tau$, where $\tau$ is the maximally mixed state. We determine the key rate as a function of $p$, i.e. we determine the amount of key that can safely be generated from any potentially dishonest implementation that produces the same statistics as the honest implementation with noise level $p$. To this end, for every value of $p$ we first determine the statistics produced by an honest implementation with that noise level. We then choose a collective attack bound and parameters for Theorem 4 that ensure that the protocol is $\varepsilon^{\mathrm{cor}}$-correct, $\varepsilon^{\mathrm{sec}}$-secret, and $\varepsilon^{\mathrm{comp}}$-complete for that noise level and $\varepsilon^{\mathrm{cor}} = 5 \cdot 10^{-11}, \varepsilon^{\mathrm{sec}} = 10^{-9}$, and $\varepsilon^{\mathrm{comp}} = 10^{-2}$. Finally, we choose the key length to be the largest integer $l$ that satisfies the condition in Equation (12). We provide the choice of parameters in detail in Supplementary Note F and plot the resulting key rate in Fig. 1 for different numbers of rounds $n$. We again note that the choice of parameters here is largely arbitrary and not optimised as the purpose of this example is only to illustrate the use of our general framework.

## Discussion
We have introduced a proof technique for analysing the security of QKD protocols in the finite-size regime against general attacks. This technique is best understood as a general procedure for converting a security proof in the i.i.d. asymptotic setting into a finite-size security proof against general attacks. To apply our technique, one can express a protocol of interest as an instance of our template protocol in Box 1, derive a collective attack bound (either using the general numerical technique described in "Results" subsection "Collective attack bounds" or by reusing an existing analysis in the i.i.d. asymptotic setting), and apply our Theorem 4 to obtain finite-size key rates against general attacks. Unlike previous techniques, our method can be applied directly to prepare-and-measure protocols and does not depend on the dimension of the underlying Hilbert space, allowing for a simple analysis of photonic prepare-and-measure protocols.

While we have provided a simple illustrative example of applying our framework to the well-known B92 protocol (Results Subsection "Sample application: B92 protocol"), which is not amenable to

treatment with the EAT, and sketched the analysis of the BB84 decoy-state protocol (Supplementary Note G), we leave it for future work to analyse more practical protocols and optimise the bounds one can obtain for those protocols. This is especially relevant given that commercial QKD systems may become increasingly prevalent in the near future. In particular, it would be interesting to see whether our framework can be used to prove the security of the differential phase-shift[42] and coherent one-way[43] QKD protocols. These protocols (and related ones using similar ideas) are relatively practical to implement, but notoriously hard to analyse.

## Methods
### Notation
The set of states for a quantum system $A$ (with associated Hilbert space $\mathcal{H}_A$) is given by $S(A) = \{\rho \in \mathrm{Pos}(A) \mid \mathrm{Tr}[\rho] = 1\}$, where $\mathrm{Pos}(A)$ is the set of positive operators on $\mathcal{H}_A$. If $A$ is a quantum system and $X$ is a classical system with alphabet $\mathcal{X}$, we call $\rho \in S(XA)$ a cq-state and can expand it as $\rho_{XA} = \sum_{x \in \mathcal{X}} |x\rangle\langle x| \otimes \rho_{A,x}$ for subnormalised $\rho_{A,x} \in \mathrm{Pos}(A)$. For $\Omega \subset \mathcal{X}$, we define the partial and conditional states

$$\rho_{XA\wedge\Omega} = \sum_{x \in \Omega} |x\rangle\langle x| \otimes \rho_{A,x} \text{ and } \rho_{XA|\Omega} = \frac{1}{\mathrm{Pr}_\rho[\Omega]}\rho_{XA\wedge\Omega}, \tag{25}$$

where $\mathrm{Pr}_\rho[\Omega] := \mathrm{Tr}[\rho_{XA\wedge\Omega}]$. If $\Omega = \{x\}$, we also write $\rho_{XA|x}$ for $\rho_{XA|\Omega}$. The set of quantum channels from system $A$ to $A'$ is denoted as $\mathrm{CPTP}(A,A')$. The trace norm (sum of the singular values) of an operator $L$ on $\mathcal{H}_A$ is denoted as $|L|_1$.

We will deal with two different entropies, the von Neumann entropy and the min-entropy, which are defined as follows. Let $\rho_{AB} \in S(AB)$ be a quantum state. Then the conditional von Neumann entropy of $A$ conditioned on $B$ is given by

$$H(A|B)_\rho = -\mathrm{Tr}[\rho_{AB}\log\rho_{AB}] + \mathrm{Tr}[\rho_B\log\rho_B]. \tag{26}$$

For $\varepsilon \in [0,1]$, the $\varepsilon$-smoothed min-entropy of $A$ conditioned on $B$ is

$$H_{\min}^\varepsilon(A|B)_\rho = -\log\inf_{\tilde{\rho}_{AB}}\inf_{\sigma_B \in S(B)}\left\|\sigma_B^{-\frac{1}{2}}\tilde{\rho}_{AB}\sigma_B^{-\frac{1}{2}}\right\|_\infty, \tag{27}$$

where $\|\cdot\|_\infty$ denotes the spectral norm and the first infimum is taken over all states $\tilde{\rho}_{AB} \in \mathcal{B}_\varepsilon(\rho_{AB})$ in the $\varepsilon$-ball around $\rho_{AB}$ (in terms of the purified distance[44]).

### Universal hashing and randomness extraction
To check that Alice's and Bob's keys are the same, our general QKD protocol will make use of a universal hash family, and to extract a secure key from Alice's and Bob's raw data we will use a randomness extractor. Here, we briefly define what these primitives achieve. We refer to ref. 15 for a more detailed exposition and explanation of their construction.

**Definition 5.** (Universal hash family) Let $M$ be a set. A family $\mathcal{F}$ of functions from $M$ to $\{0,1\}^l$ with a probability distribution $P_\mathcal{F}$ over $\mathcal{F}$ is called a universal hash family if for any $x \neq x' \in M, \mathrm{Pr}_f[f(x) = f(x')] \leq 2^{-l}$.

**Definition 6.** (Quantum-proof strong extractor[15,45,46]) A function $\mathrm{EXT}: \{0,1\}^m \times \{0,1\}^d \to \{0,1\}^l$ is a quantum-proof strong $(k, \varepsilon_{\mathrm{EXT}})$-extractor if for any $\rho_{SE} \in \mathrm{Pos}(SE)$ with $\mathrm{Tr}[\rho] \leq 1$ (and $S$ classical with dimension $2^m$) for which $H_{\min}(S|E)_\rho \geq k$, we have

$$\left\|\mathrm{EXT}(\rho_{SE} \otimes \tau_D) - \tau_K \otimes \rho_E \otimes \tau_D\right\|_1 \leq \varepsilon_{\mathrm{EXT}}, \tag{28}$$

where $\tau_D$ and $\tau_K$ are maximally mixed states of dimension $2^d$ and $2^l$, respectively, and the map EXT acts on the classical systems $S$ and $D$. The input on system $D$ is called the seed of the extractor.

This definition of extractors makes use of the non-smoothed min-entropy $H_{min}(S|E)_\rho$. It is straightforward to modify this condition so that it only requires a lower bound on the smooth min-entropy: if EXT is a quantum-proof strong $(k, \varepsilon_{EXT})$-extractor as in Definition 7 and $\rho_{SE}$ satisfies $H^\varepsilon_{min}(S|E)_\rho \ge k$, then

$$\left\|\text{EXT}(\rho_{SE} \otimes \tau_D) - \tau_L \otimes \rho_E \otimes \tau_D\right\|_1 \le \varepsilon_{EXT} + 4\varepsilon. \quad (29)$$

To see that this is the case, note that $H^\varepsilon_{min}(S|E)_\rho \ge k$ means that there exists a $\rho'$ within $\varepsilon$ purified distance of $\rho$ for which $H_{min}(S|E)_{\rho'} \ge k$. By the relation between purified distance and trace distance[44], we have $\|\rho - \rho'\|_1 \le 2\varepsilon$. Then, Equation (29) follows from the triangle inequality and because applying the map EXT cannot increase the trace distance.

For the purposes of QKD, a simple construction based on two-universal hashing[15] provides sufficiently good parameters. We also note that more involved constructions exist that require shorter seeds, but this if typically not a concern for QKD applications (see e.g.[46] for a very efficient example using Trevisan's extractor).

**Lemma 7.** (ref. [15]) There exist quantum-proof strong $(k, \varepsilon_{EXT})$-extractors EXT: $\{0,1\}^m \times \{0,1\}^d \to \{0,1\}^l$ for $d = m$ and $l \le k - 2\log(1/\varepsilon_{EXT})$.

## Generalised entropy accumulation

In this section, we introduce the GEAT from ref. [36]. Most of this section is taken directly from[36] and we refer to the introduction of that paper for a more detailed description of the setting and how it compares to the EAT[17]. Consider a sequence of channels $\mathcal{M}_i \in \text{CPTP}(R_{i-1}E_{i-1}, C_iA_iR_iE_i)$ for $i \in \{1, ..., n\}$, where $C_i$ are classical systems with common alphabet $\mathcal{C}$. In the context of cryptographic protocols, one should think of $E_i$ as Eve's side information after the $i$-th round, $R_i$ as some internal system of a device, $A_i$ as the protocol's output in the $i$-th round, and $C_i$ as classical statistics that determine whether the protocol aborts (e.g. by checking the number of rounds on which $A_i$ does not satisfy a certain property). For all results in this paper, $R_i$ can be chosen to be trivial. However, for (semi-)device-independent applications, the systems $R_i$ are important because they can be used to describe the internal memory of the untrusted devices. As this is an interesting direction for future work, we state the theorem in full generality here.

We require that these channels $\mathcal{M}_i$ satisfy the following condition: defining $\mathcal{M}'_i = \text{Tr}_{C_i} \circ \mathcal{M}_i$ (where $\text{Tr}_{C_i}$ is the partial trace over system $C_i$ and $\circ$ is the composition of channels), there exists a channel $\mathcal{T} \in \text{CPTP}(A^nE_n, C^nA^nE_n)$ such that $\mathcal{M}_n \circ \cdots \circ \mathcal{M}_1 = \mathcal{T} \circ \mathcal{M}'_n \circ \cdots \circ \mathcal{M}'_1$ and $\mathcal{T}$ has the form

$$\mathcal{T}(\omega_{A^nE_n}) = \sum_{y \in \mathcal{Y}, z \in \mathcal{Z}} \left(\Pi^{(y)}_{A^n} \otimes \Pi^{(z)}_{E_n}\right) \omega_{A^nE_n} \left(\Pi^{(y)}_{A^n} \otimes \Pi^{(z)}_{E_n}\right) \otimes |r(y,z)\rangle\langle r(y,z)|_{C^n}, \quad (30)$$

where $\{\Pi^{(y)}_{A^n}\}$ and $\{\Pi^{(z)}_{E_n}\}$ are families of mutually orthogonal projectors on $A_i$ and $E_i$, and $r : \mathcal{Y} \times \mathcal{Z} \to \mathcal{C}$ is a deterministic function. Intuitively, this condition says that the classical statistics can be reconstructed "in a projective way" from systems $A^n$ and $E_n$ at the end of the protocol. In particular, this requirement is always satisfied if the statistics are computed from classical information contained in $A^n$ and $E_n$, which is the case for the applications in this paper. We note that the statistics are still generated in a round-by-round manner; Eq. (30) merely asserts that they could be reconstructed from the final state.

Let $\mathbb{P}$ be the set of probability distributions on the alphabet $\mathcal{C}$ of $C_i$, and let $\bar{E}_{i-1}$ be a system isomorphic to $R_{i-1}E_{i-1}$. For any $q \in \mathbb{P}$ we define the set of states

$$\Sigma_i(q) = \left\{\nu_{C_iA_iR_iE_i\bar{E}_{i-1}} = \mathcal{M}_i(\omega_{R_{i-1}E_{i-1}\bar{E}_{i-1}}) | \omega \in S(R_{i-1}E_{i-1}\bar{E}_{i-1}) \text{ and } \nu_{C_i} = q \right\}, \quad (31)$$

where $\nu_{C_i}$ denotes the probability distribution over $\mathcal{C}$ with the probabilities given by $\text{Pr}[c] = \langle c|\nu_{C_i}|c\rangle$. In other words, $\Sigma_i(q)$ is the set of states

that can be produced at the output of the channel $\mathcal{M}_i$ and whose reduced state on $C_i$ is equal to the probability distribution $q$.

**Definition 8.** A function $f : \mathbb{P} \to \mathbb{R}$ is called a min-tradeoff function for $\{\mathcal{M}_i\}$ if it satisfies

$$f(q) \le \min_{\nu \in \Sigma_i(q)} H(A_i|E_i\bar{E}_{i-1})_\nu \quad \forall i = 1, ..., n. \quad (32)$$

Note that if $\Sigma_i(q) = \emptyset$, then $f(q)$ can be chosen arbitrarily.

Our result will depend on some simple properties of the tradeoff function, namely the maximum and minimum of $f$, the minimum of $f$ over valid distributions, and the maximum variance of $f$:

$$\text{Max}(f) := \max_{q \in \mathbb{P}} f(q), \quad (33)$$

$$\text{Min}(f) := \min_{q \in \mathbb{P}} f(q), \quad (34)$$

$$\text{Min}_\Sigma(f) := \min_{q:\Sigma(q) \ne \emptyset} f(q), \quad (35)$$

$$\text{Var}(f) := \max_{q:\Sigma(q) \ne \emptyset} \sum_{x \in \mathcal{C}} q(x) f(\delta_x)^2 - \left(\sum_{x \in \mathcal{C}} q(x) f(\delta_x)\right)^2, \quad (36)$$

where $\Sigma(q) = \bigcup_i \Sigma_i(q)$ and $\delta_x$ is the distribution with all the weight on element $x$. We write $\text{freq}(C^n)$ for the distribution on $\mathcal{C}$ defined by $\text{freq}(C^n)(c) = \frac{|\{i \in \{1,...,n\}: C_i = c\}|}{n}$. We also recall that in this context, an event $\Omega$ is defined by a subset of $\mathcal{C}^n$, and for a state $\rho_{C^nA^nE_nR_n}$ we write $\text{Pr}_\rho[\Omega] = \sum_{c^n \in \Omega} \text{Tr}\left[\rho_{A^n_1E_nR_n,c^n}\right]$ for the probability of the event $\Omega$ and

$$\rho_{C^nA^nE_nR_n|\Omega} = \frac{1}{\text{Pr}_\rho[\Omega]} \sum_{c^n \in \Omega} |c^n\rangle\langle c^n|_{C^n} \otimes \rho_{A^nE_nR_n,c^n} \quad (37)$$

for the state conditioned on $\Omega$. With this, we can finally state the GEAT of[36].

**Theorem 9.** (GEAT[36]) Consider a sequence of channels $\mathcal{M}_i \in \text{CPTP}(R_{i-1}E_{i-1}, C_iA_iR_iE_i)$ for $i \in \{1, ..., n\}$, where $C_i$ are classical systems with common alphabet $\mathcal{C}$ and the sequence $\{\mathcal{M}_i\}$ satisfies Equation (30) and the following no-signalling condition: for each $\mathcal{M}_i$, there exists a channel $\mathcal{R}_i \in \text{CPTP}(E_{i-1}, E_i)$ such that $\text{Tr}_{A_iR_iC_i} \circ \mathcal{M}_i = \mathcal{R}_i \circ \text{Tr}_{R_{i-1}}$. Let $\varepsilon \in (0,1), \alpha \in (1, 3/2), \Omega \subset \mathcal{C}^n, \rho_{R_0E_0} \in S(R_0E_0)$, and $f$ be an affine min-tradeoff function with $h = \min_{c^n \in \Omega} f(\text{freq}(c^n))$. Then,

$$H^\varepsilon_{min}(A^n|E_n)_{\mathcal{M}_n \circ \cdots \circ \mathcal{M}_1(\rho_{R_0E_0})_{|\Omega}} \ge nh - n\frac{\alpha - 1}{2 - \alpha}\frac{\ln(2)}{2}V^2$$
$$- \frac{g(\varepsilon) + \alpha\log(1/\text{Pr}_{\rho^n}[\Omega])}{\alpha - 1} - n\left(\frac{\alpha - 1}{2 - \alpha}\right)^2 K'(\alpha), \quad (38)$$

where $\text{Pr}[\Omega]$ is the probability of observing event $\Omega$, and

$$g(\varepsilon) = -\log(1 - \sqrt{1 - \varepsilon^2}) \le \log(2/\varepsilon^2), \quad (39)$$

$$V = \log(2d_A^2 + 1) + \sqrt{2 + \text{Var}(f)}, \quad (40)$$

$$K'(\alpha) = \frac{(2 - \alpha)^3}{6(3 - 2\alpha)^3 \ln 2} 2^{\frac{\alpha - 1}{2 - \alpha}(2\log d_A + \text{Max}(f) - \text{Min}_\Sigma(f))}$$
$$\ln^3\left(2^{2\log d_A + \text{Max}(f) - \text{Min}_\Sigma(f)} + e^2\right), \quad (41)$$

with $d_A = \max_i \dim(A_i)$.

We briefly comment on the main differences between the GEAT as stated above and the EAT from[17]. The GEAT deals with a sequence of channels $\mathcal{M}_i \in \text{CPTP}(R_{i-1}E_{i-1}, C_iR_iE_i)$ that can update both the internal memory register $R_i$ and the side information register $E_i$ (subject to the no-signalling condition), i.e. change these states to e.g. incorporate additional side information obtained in the protocol or account for measurements performed in response to the user's input. In contrast, the EAT does not allow the side information register to be updated. More formally, the EAT deals with channels $\mathcal{M}'_i \in \text{CPTP}(R_{i-1}, C_iA_iR_iI_i)$, where $I_i$ is side information produced in each round that cannot be updated in the future. The final side information at the end of such a process is $EI^n$, where $E$ can be any additional side information from the initial state of the process that was never updated during the process. If the side information registers $I_i$ satisfy the Markov condition $A^{i-1} \leftrightarrow I^{i-1}E \leftrightarrow I_i$ (see ref. [17] for a more detailed explanation), then the EAT gives a lower bound on $H^\varepsilon_{\min}(A^n|I^nE)_{\mathcal{M}'_n \circ \cdots \circ \mathcal{M}'_1(\rho_{R_0})_{|\Omega}}$ similar to the one in Theorem 9.

We can now see at a high level why the EAT cannot be used to deal with prepare-and-measure protocols directly: in a prepare-and-measure protocol, the adversary Eve intercepts the quantum state sent from Alice to Bob in each round and updates her side information based on that. Therefore, any technique used to deal with such protocols must allow for the side information to be updated like in the GEAT; the more restrictive scenario considered in the EAT does not capture this kind of protocol.

We also note that the GEAT is strictly more general than the EAT (see [ref. 36, Section 1] for a proof). Hence, any application that can be treated with the EAT can also be treated with the GEAT (up to some very minor loss in second-order parameters), and the resulting proofs are often much more straightforward; see [ref. 36, Section 5.2] for an example.

## Proof of main theorem

In this section, we prove our main result, Theorem 4, i.e. we show that the protocol in Box 1 is correct and secret.

**Proof of Theorem 4.** For the correctness statement, we need to show that $\Pr[K \neq \hat{K} \wedge \text{not abort}] \leq \varepsilon_{\text{KV}}$. To see that this is the case, we note that due to the check in Step (5), the protocol not aborting implies that $\text{HASH}(S^n) = \text{HASH}(\hat{S}^n)$. Furthermore, from Step (7) we see that $K \neq \hat{K}$ implies that $S^n \neq \hat{S}^n$. Therefore, it suffices to show that

$$\Pr\left[S^n \neq \hat{S}^n \wedge \text{HASH}(S^n) = \text{HASH}(\hat{S}^n)\right] \leq \varepsilon_{\text{KV}}. \tag{42}$$

Since Alice chooses the function HASH at random from a universal hash family, this follows directly from Definition 5 and completes the correctness proof.

The remainder of the proof will be concerned with the secrecy condition. As explained in "Results" subsection "Modelling Eve's attack", assuming Condition 1 we can model a general attack by a sequence of channels

$$\mathcal{A}_i : E'_{i-1}Q_i \to E'_iQ_i. \tag{43}$$

Alice, Bob, and Eve's joint final state at the end of the protocol therefore contains systems

$$U^nV^nI^nS^n\hat{S}^n\hat{C}^nK\hat{K}E'_nE'. \tag{44}$$

Here, $E'_n$ is Eve's system after using the maps $\mathcal{A}_1, \ldots, \mathcal{A}_n$, $E'$ stores the additional classical information published after Step (4), i.e., the error correction information EC, a description of the hash function HASH, the hash value $\text{HASH}(S^n)$, and the seed $\mu$, and the other systems are labelled as in Box 1. This means that Eve's full side information is given

by $I^nE'_nE'$. Throughout the proof, we will denote the final state at the end of the protocol by $\rho_{U^nV^nI^nS^n\hat{S}^n\hat{C}^nK\hat{K}E'_nE'}$.

By Definition 3, we need to show that

$$\left\| \rho_{KI^nE'_nE'\wedge\Omega} - \tau_K \otimes \rho_{I^nE'_nE'\wedge\Omega} \right\|_1 \leq \max\{\varepsilon_{\text{PA}} + 4\,\varepsilon_s, 2\,\varepsilon_a\} + 2\,\varepsilon_{\text{KV}}, \tag{45}$$

where $\Omega$ is the event that the protocol does not abort and $\tau_K$ is the maximally mixed state on system $K$ of dimension $|K| = 2^l$. Since the protocol's final state arises by application of a strong extractor in Step (7), we can reduce Eq. (45) to an entropic statement. This step requires careful technical treatment because the statistical check in Step (6) uses the systems $\hat{C}^n$, which are computed from $\hat{S}^n$. However, $\hat{S}^n$ is Bob's guess for Alice's string $S^n$ and depends on the global error correction information EC, i.e., it cannot be generated in a round-by-round manner as required for the GEAT. The intuition for circumventing this issue is as follows: if $\hat{S}^n \neq S^n$, then the protocol is likely to abort anyway because of Step (5); on the other hand, if $\hat{S}^n = S^n$, then we can replace $\hat{S}^n$ by $S^n$, and the latter is generated in a round-by-round manner. Following this intuition, we can show that the entropy bound in Claim 10 implies Theorem 4. We give a formal proof of this step in Supplementary Note A and continue here with proving the required entropy bound. We also note that for protocols that include a separate parameter estimation step rather than using Bob's guess for Alice's raw key, Claim 10 implies Theorem 4 almost immediately. □

**Claim 10.** Let $\Omega_C$ be the event that $\text{CA}(\text{freq}(C^n)) \geq k_{\text{CA}}$ (i.e. the statistical check (Step (6)) passes using the values $C^n$). Continuing with the notation from before, for any $\alpha \in (1, 3/2)$:

$$H^{\varepsilon_s}_{\min}(S^n|I^nC^nE'_n)_{\rho_{|\Omega_C}} \geq nk_{\text{CA}} - n\frac{\alpha-1}{2-\alpha}\frac{\ln(2)}{2}V^2 - \frac{g(\varepsilon_s) + \alpha\log(1/\Pr[\Omega_C])}{\alpha-1} - n\left(\frac{\alpha-1}{2-\alpha}\right)^2 K'(\alpha), \tag{46}$$

with $g(\varepsilon_s)$, $V$, and $K'(\alpha)$ as in Theorem 9.

**Proof.** To make use of the GEAT, we need to write $\rho_{S^nI^nC^nE'_n|\Omega_C}$ as the result of a sequential application of a quantum channel. For this we fix an attack $\mathcal{A}_1, \ldots, \mathcal{A}_n$ and define

$$\mathcal{M}_i : E'_{i-1} \to S_iI_iC_iE'_i \tag{47}$$

as the following channel: given a quantum system $\omega_{E'_{i-1}}$,
(i)   create the state $\psi_{U_iQ_i}$ (defined in Step (1) of Box 1),
(ii)   apply the attack map $\mathcal{A}_i : Q_iE'_{i-1} \to Q_iE'_i$ to $\psi_{U_iQ_i} \otimes \omega_{E'_{i-1}}$,
(iii)   measure $\{N^{(v)}\}_{v \in \mathcal{V}}$ on system $Q_i$ and store the result in register $V_i$,
(iv)   set $I_i = \text{PD}(U_i, V_i)$,
(v)   set $S_i = \text{RK}(U_i, I_i)$,
(vi)   set $C_i = \text{EV}(V_i, I_i, S_i)$,
(vii)   trace out registers $U_i$ and $V_i$. Comparing the steps of the protocol and Supplementary Eq. (1) with this definition of $\mathcal{M}_i$, we see that the marginal of $\rho$ on systems $S^nI^nC^nE'_n$ is the same as the output of the maps $\mathcal{M}_i$:

$$\rho_{S^nI^nC^nE'_n} = \mathcal{M}_n \circ \cdots \circ \mathcal{M}_1(\omega_{E'_0}), \tag{48}$$

where $\omega_{E_0}$ is the initial state of Eve's side information (which can be chosen to be trivial without loss of generality as explained in Results Subsection "Modelling Eve's attack"). If we define the systems $E_i = I^iC^iE'_i$, then by suitable tensoring with the identity map and copying the register $C_i$ we can view $\mathcal{M}_i$ as a map

$$\tilde{\mathcal{M}}_i : E_{i-1} \to S_iE_iC_i. \tag{49}$$

With this we can also express the final state (which technically now includes two copies of $C^n$, one explicit and one part of $E_n$) as

$$\rho_{S^n E_n C^n} = \tilde{\mathcal{M}}_n \circ \cdots \circ \tilde{\mathcal{M}}_1(\omega_{E_0}). \tag{50}$$

With this notation, the entropy on the l.h.s. of Equation (46) can be written as

$$H_{\min}^{\varepsilon_s}(S^n | I^n C^n E'_n)_{\rho_{|\Omega_C}} = H_{\min}^{\varepsilon_s}(S^n | E_n)_{\tilde{\mathcal{M}}_n \circ \cdots \circ \tilde{\mathcal{M}}_1(\omega_{E_0})_{|\Omega_C}}. \tag{51}$$

We want to apply Theorem 9 to derive the desired lower bound in Eq. (46). For this, we first need to check that the required conditions on the maps $\tilde{\mathcal{M}}_i$ are satisfied. The condition in Eq. (30) is clearly satisfied as the systems $C_i$ are themselves included in the conditioning system $E'_n$. The non-signalling condition in Theorem 9 is also trivially satisfied in this case since there is no system $R_i$.

We now need to argue that the collective attack bound CA : $\mathbb{P}(\mathcal{C}) \to \mathbb{R}$ used as an argument in Box 1 is a min-tradeoff function for the maps $\{\tilde{\mathcal{M}}_i\}$. By Definition 8, we need to show that for any $i$, attack $\mathcal{A}_i : Q_i E'_{i-1} \to Q_i E'_i$ (in the definition of $\tilde{\mathcal{M}}_i$, see Step (ii)), and state $\omega_{\bar{E}_{i-1}\bar{E}_{i-1}}^{i-1}$ (where $\bar{E}_{i-1} \equiv E_{i-1}$), the following holds:

$$\text{CA}(\tilde{\mathcal{M}}_i(\omega^{i-1})_{C_i}) \le H(S_i | E_i \bar{E}_{i-1})_{\tilde{\mathcal{M}}_i(\omega^{i-1})}. \tag{52}$$

For the rest of the proof, we fix an arbitrary choice of $i, \omega^{i-1}$, and $\mathcal{A}_i$. To relate Eq. (52) to the definition of collective attack bounds ("Definition 2"), we construct a collective attack $\mathcal{A}' : Q_i \to Q_i E_i \bar{E}_{i-1}$ such that

$$\tilde{\mathcal{M}}_i(\omega^{i-1})_{S_i C_i E_i \bar{E}_{i-1}} = \nu_{S_i C_i E_i \bar{E}_i}, \tag{53}$$

where $\nu$ is defined as in Definition 2, i.e. $\nu$ is the state produced by running a single round of the protocol in Box 1 with the attack $\mathcal{A}'$. Of course, $\mathcal{A}'$ will depend on $i, \omega^{i-1}$, and $\mathcal{A}_i$. This is not a problem since Definition 2 holds for any collective attack, i.e., to show that Eq. (52) holds for any $i, \omega^{i-1}$, and $\mathcal{A}_i$, we can first fix an arbitrary choice, construct a "custom" collective attack that shows Eq. (52) for that choice, and then apply the condition in Definition 2 to that choice.

It is easy to check that Eq. (53) is satisfied for the following choice of $\mathcal{A}'$: given a state $\sigma_Q, \mathcal{A}'$ first creates the (fixed) state $\omega_{E_{i-1}\bar{E}_{i-1}}^{i-1}$ and then applies the (fixed) attack $\mathcal{A}_i$ to $\sigma_Q \otimes \omega_{E_{i-1}\bar{E}_{i-1}}^{i-1}$ (with $Q_i = Q$).

Then, since CA is a collective attack bound, Eq. (52) follows from Definition 2:

$$\text{CA}(\tilde{\mathcal{M}}_i(\omega^{i-1})_{C_i}) = \text{CA}(\nu_{C_i}) \le H(S_i | E_i \bar{E}_{i-1} C_i)_\nu = H(S_i | E_i \bar{E}_{i-1})_{\tilde{\mathcal{M}}_i(\omega^{i-1})}. \tag{54}$$

Compared to Definition 2, we have dropped the explicit conditioning on $I := I_i$ since $I_i$ is already part of $E_i$, and in the last equality we can drop $C_i$ since it is also part of $E_i$.

This means that the function CA is a min-tradeoff function for the protocol in Box 1. By definition, for any $c^n \in \Omega_C$, $\text{CA}(\text{freq}(c^n)) \ge k_{CA}$. Hence, Claim 10 follows by applying Theorem 9. □

Having proved correctness and secrecy, we turn our attention to the completeness of the protocol in Box 1, i.e. we need to bound the probability that the protocol aborts when Eve does not interfere in the protocol, but the channel between Alice and Bob may be noisy. In the protocol, Alice sends a quantum system $Q$ to Bob. If the channel connecting Alice and Bob is noisy, instead of Alice's and Bob's joint state in each round being $\psi_{UQ}$, the joint state is $\mathcal{N}(\psi_{UQ})$ for some channel $\mathcal{N} : Q \to Q$. This channel $\mathcal{N}$ describes the noise model for Box 1. Note that the channel $\mathcal{N}$ is not something that needs to be added explicitly to the description of Box 1: formally, $\mathcal{N}$ can be viewed as Eve's attack, i.e. we can model the implementation of the protocol in Box 1 with a noisy channel and honest Eve by saying that Eve's attack is

described by $\mathcal{N}$. This also means that when we proved correctness and secrecy, we only needed to prove this for any behaviour of Eve, not any noise model, since the noise model can be included in Eve's actions.

For a given noise model $\mathcal{N}$, we need to choose the length of the error correction string $\lambda_{EC}$ to be sufficiently long such that Bob's guess $\hat{S}^n$ for Alice's raw key $S^n$ is correct with high probability, and as a consequence the check in Step (5) passes. Furthermore, we need to choose the threshold $k_{CA}$ to be sufficiently low that an honest noisy state passes Step (6) with high probability. The precise choice of parameters can be worked out using the properties of the error correcting code in Step (4) and statistical tail bounds for Step (6). We provide the details in Supplementary Note B.

### Deriving collective attack bounds

Our main result, Theorem 4, turns an affine collective attack bound (defined in "Definition 2") into a security statement against general attacks. Therefore, the main step one has to perform to use our framework is finding such an affine collective attack bound for a protocol of interest. In this section, we give a numerical method for finding collective attack bounds for the protocol in Box 1 based on ideas from refs. 7,47. Combined with Theorem 4, this means that the problem of finding key rate bounds against general attacks for any instance of the protocol in Box 1 is reduced to a numerical computation.

We begin by noting that we can rewrite the condition Eq. (8) from "Definition 2" as follows: for any probability distribution $\nu_C^* \in \mathbb{P}(\mathcal{C})$ we require that

$$\inf_{\nu \text{ s.t. } \nu_C = \nu_C^*} H(S | IEC)_\nu \ge \text{CA}(\nu_C^*), \tag{55}$$

where the infimum is over all states $\nu$ that can result from a collective attack and have statistics $\nu_C^*$ (and the infimum is infinite if there is no such state). In the language of the GEAT, a collective attack bound essentially is a min-tradeoff function for a certain sequence of maps associated with Box 1. More details on how a collective attack bound serves as a min-tradeoff function can be found in the proof of Claim 10.

Since we are interested in an affine lower bound, we write the probability distribution $\nu_C$ as a probability vector $\vec{\nu}_C$ and, following[12,48], make the ansatz

$$\text{CA}(\vec{\nu}_C) = \vec{\lambda} \cdot \vec{\nu}_C + c_{\vec{\lambda}} \tag{56}$$

for some vector $\vec{\lambda}$ of the same dimension as $\vec{\nu}_C$ and a constant $c_{\vec{\lambda}}$. We treat $\vec{\lambda}$ as a parameter that will be chosen heuristically. For example, one can choose $\vec{\lambda}$ by numerically estimating the gradient of the function $\nu_C' \mapsto \inf_{\nu \text{ s.t. } \nu_C = \nu_C'} H(S | IEC)_\nu$ around a particular choice of classical statistics $\nu_C^*$ that has been observed in an experimental realisation of the protocol, although this choice is not necessarily optimal and $\vec{\lambda}$ should be numerically optimised if one wants to obtain the best possible key rates.

Having chosen $\vec{\lambda}$ heuristically, we need to compute a value of $c_{\vec{\lambda}}$ that ensures that $\vec{\lambda} \cdot \vec{\nu}_C + c_{\vec{\lambda}}$ is a valid min-tradeoff function. Inserting our ansatz into Equation (8), we see that for any fixed $\vec{\lambda}$, a valid choice of $c_{\vec{\lambda}}$ is one that satisfies

$$c_{\vec{\lambda}} \le \inf_\nu H(S | IEC)_\nu - \vec{\lambda} \cdot \vec{\nu}_C. \tag{57}$$

The infimum here is taken over the states $\nu$ described in "Definition 2". To avoid confusion, we emphasise that the infimum here is taken over

all such states $\nu$, not just ones with a specific classical distribution $\nu_C^*$ as considered in Eq. (55). As explained in ref. 12, one can view the optimisation in Eq. (57) as arising from the Lagrange dual of Eq. (55), but we will not make use of this relation here explicitly.

To tackle this optimisation problem, we consider an entanglement-based version of the protocol in Box 1 using the source-replacement scheme explained in ref. 6. As explained in the introduction, switching to an entanglement-based version of a prepare-and-measure protocol generally requires introducing "artificial" constraints on Eve's actions. These artificial constraints are troublesome when applying the EAT to the entanglement-based version, but here we take a different approach: we only use the entanglement-based version to derive a collective attack bound (for which the artificial constraints do not present a problem). This collective attack bound also applies to the original prepare-and-measure protocol and in Theorem 4 we apply the EAT with this collective attack bound to the prepare-and-measure protocol directly. We emphasise that the method for deriving a collective attack bound and our Theorem 4 are entirely independent: Theorem 4 does not depend on how the collective attack bound was derived and does not make use of an entanglement-based protocol itself.

In Box 1 Alice prepares the state

$$\psi_{UQ} = \sum_u p(u)|u\rangle\langle u| \otimes |\psi\rangle\langle\psi|_{Q|u} \qquad (58)$$

and sends system $Q$ to Bob. It is clear that Alice could equivalently prepare the state

$$|\bar{\psi}\rangle_{UQ} = \sum_u \sqrt{p(u)}|u\rangle_P \otimes |\psi\rangle_{Q|u'} \qquad (59)$$

send system $Q$ to Bob, and only afterwards measure her own system $P$ in the computational basis, storing the outcome in system $U$. Eve would now apply her collective attack $\mathcal{A}: Q \to QE$ to system $Q$ of $\bar{\psi}$, so the state after Eve's attack would be $\tilde{\psi}_{PQE}$. We can replace this attack by giving Eve the ability to prepare a state $\hat{\psi}_{PQE}$ directly and distribute $P$ and $Q$ to Alice and Bob, respectively. This kind of attack clearly gives Eve more power. In fact, it gives Eve too much power: in order to still obtain a good key rate, we need to enforce the additional constraint that Alice's marginal of the state $\hat{\psi}$ is the same as her marginal of the state $\tilde{\psi}$ she would have prepared herself, i.e. $\hat{\psi}_P = \tilde{\psi}_P$. It is easy to see that even with this additional constraint, this latter kind of attack is still at least as general as any collective attack on the prepare-and-measure protocol described before. Note that the condition $\hat{\psi}_A = \tilde{\psi}_A$ is not a physical constraint that Alice checks in an actual protocol, but rather the aforementioned additional artificial constraint. Nonetheless, we can impose this artificial constraint on the optimisation problem used to calculate the collective attack bound.

For a fixed instance of Box 1, we can now view the state $\nu$ in "Definition 2" as a function of $\hat{\psi}_{PQE}$:

$$\nu_{ESIC}(\hat{\psi}) = \sum_{u,v} \text{Tr}_{PQ}\Big[|u\rangle\langle u|_P \otimes N_Q^{(v)}\hat{\psi}_{PQE}\Big] \otimes |\text{RK}(u,i)\rangle\langle|_S \otimes |\text{PD}(u,v)\rangle\langle|_I \otimes |\text{EV}(v,i)\rangle\langle|_C. \qquad (60)$$

Here, $|\text{RK}(u,\text{PD}(u,v))\rangle\langle|$ is shorthand for the projector $|\text{RK}(u,\text{PD}(u,v))\rangle\langle\text{RK}(u,\text{PD}(u,v))|$ and $i$ is shorthand for $\text{PD}(u,v)$. We can therefore write the optimisation problem from Equation (57) as

$$\inf_{\psi_{PQE}} H(S|IEC)_\nu - \overrightarrow{\lambda} \cdot \overrightarrow{\nu}_C \qquad (61)$$

$$\text{s.t. } \hat{\psi}_{PQE} \geq 0, \quad \text{Tr}\Big[\hat{\psi}_{PQE}\Big] = 1, \quad \hat{\psi}_P = \tilde{\psi}_P, \qquad (62)$$

where $\nu = \nu(\hat{\psi})$, and without loss of generality we can restrict the optimisation to pure states on $PQE$ with $E \equiv PQ$.

A lot of work in QKD has been focused on numerical methods for this kind of optimisation problem (see e.g. refs. 6,7,13,49,50). The key difficulty is that we need a lower bound on the infimum of a concave function $H(S|IEC)_{\nu(\hat{\psi})}$. Here we use a method from refs. 7,47 to turn this optimisation problem into a convex one. As a first step, we observe that in the definition of $\nu$ we can incorporate the classical functions RK, PD, and EV into Alice's and Bob's measurements by defining

$$M_{PQ}^{(s,i,c)} = \sum_{\substack{u,v: \\ \begin{cases} \text{RK}(u,i)=s, \\ \text{PD}(u,v)=i, \\ \text{EV}(v,i,s)=c \end{cases}}} |u\rangle\langle u|_P \otimes N_Q^{(v)}. \qquad (63)$$

Then, we can write $\nu_{ESIC}$ as

$$\nu = \sum_{s,i,c} \text{Tr}_{PQ}\Big[M_{PQ}^{(s,i,c)}\hat{\psi}_{PQE}\Big] \otimes |s\rangle\langle s|_S \otimes |i\rangle\langle i|_I \otimes |c\rangle\langle c|_C. \qquad (64)$$

Remembering that we can assume that $\psi_{PQE}$ is pure, we now define the pure state

$$|\nu^1\rangle = \sum_{s,i,c} \sqrt{M_{PQ}^{(s,i,c)}}|\hat{\psi}\rangle_{PQE}|s\rangle_S|i\rangle_I|i\rangle_{I'}|c\rangle_C|c\rangle_{C'}. \qquad (65)$$

We observe that

$$\nu_{EIC} = \nu_{EIC}^1. \qquad (66)$$

Following the proof of [ref. 47, Theorem 1], a direct calculation shows that

$$H(S|IEC)_\nu = D\Big(\nu_{PQSIC}^1 \Big\| \mathcal{P}_S(\nu_{PQSIC}^1)\Big) \qquad (67)$$

where $\mathcal{P}_S$ is the pinching map $\mathcal{P}_S(\nu^1) = \sum_{s \in \mathcal{S}}|s\rangle\langle s|_S \nu^1|s\rangle\langle s|_S$. We can view $\nu_{PQSIC}^1$ as a linear function of $\hat{\psi}_{PQ}$:

$$\nu_{PQSIC}^1(\hat{\psi}_{PQ}) = \sum_{s,s',i,c} \sqrt{M_{PQ}^{(s,i,c)}}\hat{\psi}_{PQ}\sqrt{M_{PQ}^{(s',i,c)}} \otimes |s\rangle\langle s'|_S \otimes |i\rangle\langle i|_I \otimes |c\rangle\langle c|_C, \qquad (68)$$

Furthermore, the relative entropy is jointly convex. Therefore, for a given $\overrightarrow{\lambda}$, a valid choice for $c_{\overrightarrow{\lambda}}$ can be found by solving the following convex optimisation problem:

$$c_{\overrightarrow{\lambda}} = \inf_{\psi_{PQ}} D\Big(\nu_{PQSIC}^1 \| \mathcal{P}_S(\nu_{PQSIC}^1)\Big) - \overrightarrow{\lambda} \cdot \overrightarrow{\nu}_C \qquad (69)$$

$$\text{s.t. } \hat{\psi}_{PQ} \geq 0, \quad \text{Tr}\Big[\hat{\psi}_{PQ}\Big] = 1, \quad \hat{\psi}_P = \tilde{\psi}_P, \qquad (70)$$

where $\nu_{PQSIC}^1$ and $\nu_C$ are linear functions of $\hat{\psi}_{PQ}$. To solve this optimisation problem, we can use standard techniques from convex optimisation. In particular, in refs. 41,51,52 techniques have been developed to bound the relative entropy from below by a sequence of semidefinite programs (SDPs). These SDPs can then be solved using standard SDP solvers, and the solution to the dual SDP provides a certified lower bound. Alternatively, one can also turn any feasible choice of $\hat{\psi}_{PQ}$ (ideally close to the optimal attack) into a certified lower bound using the techniques from refs. 6,7.

We note that many protocols have additional structure that allow the optimisation problem in Eq. (70) to be simplified before tackling it numerically. Additionally, if the map EV from Box 1 has a particular

structure that distinguishes between "test rounds", in which Alice and Bob use their measurement outcomes to check whether Eve tampered with the protocol, and "data rounds", in which Alice and Bob generate the raw data for their key, the derivation of a collective attack bound can be further simplified. We refer to [ref. 53, Section V.A] for a detailed explanation of this method and to Results Subsection "Sample application: B92 protocol" for an example of its use in our context.

## Data availability
No experimental data was collected as part of this work.

## Code availability
Code for reproducing Fig. 1 is available from the authors upon request.

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

## Acknowledgements
We thank Rotem Arnon-Friedman, Omar Fawzi, Marcus Haberland, Christoph Pacher, Joseph M. Renes, Martin Sandfuchs, and David Sutter for helpful discussions. We are especially grateful to Ernest Tan for helpful explanations regarding numerical methods for computing collective attack bounds. Both authors were supported by the National Centres of Competence in Research (NCCRs) QSIT and SwissMAP (both funded by the Swiss National Science Foundation), the Air Force Office of Scientific Research (AFOSR) via project No. FA9550-19-1-0202, the SNSF project No. 200021_188541 and the QuantERA project eDICT.

## Author contributions
T.M. developed the proofs and wrote the manuscript with input from R.R.; R.R. guided and supervised the project.

## Competing interests
The authors declare no competing interests.
