## [Peer Review File · Nature Communications]

Security of quantum key distribution from generalised entropy accumulationREVIEWER COMMENTS

Reviewer #1 (Remarks to the Author):

In the manuscript submitted, the authors aim to provide a general method of analysing the security of quantum key distribution (QKD) against general attacks in the finite-size regime. They defined a general prepare-and-measure device-dependent QKD protocol framework, which encompasses many of the QKD protocols of interest to the community. The security of this general protocol framework is then analysed using the generalised entropy accumulation theorem (EAT), which reduces the analysis to security against a collective attack. Security against collective attacks is generally much easier to analyse, but for the unfamiliar, a numerical method to compute the collective attack bound (from earlier work) is provided. Their main result, presented in theorem II.4, provides a simple key rate formula of QKD secure against general attacks in the finite-size regime. As an application, they demonstrated the security of the B92 QKD protocol. Instructions on how to apply generalised EAT to entanglement based protocols and decoy state BB84 are also provided in the supplementary information.

The manuscript presents an important tool, in the form of theorem II.4, for researchers in the QKD community and beyond to perform security analysis of QKD protocols by simply fitting their protocols to the protocol framework and applying the key rate formula (Eqn II.2) along with the solving for collective attack bound (for instance using Eqn IV.14). This provides an avenue for the community, especially experimentalists and those working to implement the technology, to perform security analyses of their QKD protocols, without requiring deep understanding into the underlying generalised EAT, which is highly theoretical in nature. Interestingly, the tool provided is easier to utilise than previous results with EAT, where one must convert protocols into an entanglement-based protocol, which may not be easy to do, and applies to protocols with unbounded dimensional systems, which past techniques such as the quantum de Finetti theorem cannot manage. One limitation of the proposed analysis, as mentioned in condition II.1, is the requirement that Eve can only possess one system at a time, which limits its applicability for certain applications. A workaround is briefly mentioned that can restore the analysis's validity by sacrificing the key rate.

Overall, the paper has no major issues. As far as I can tell, the claims are correct and the results presented here are interesting and useful for the community. Below are some minor issues and comments:

1. In the first line of the main text, the authors mention “authentic classical channel”. Do they mean an “authenticated classical channel”?

2. In the first page (last paragraph of first column), the authors mentioned some general numerical techniques for analysing security against collective attacks. In addition to the work that the authors have mentioned, there are other works in this topic.

(a) Wang et al, npj Quantum Information 5, 17 (2019)

(b) Primateamaja et al, PRA 99, 062332 (2019)

(c) Ref. [44] (Tan et al, npj Quantum Information 7, 158 (2021))

(d) Brown et al, arXiv: 2106.13692 (2021)

(e) Araujo et al, arXiv: 2211.05725 (2022)

(f) Hu et al, Quantum 6, 792 (2022)

3. In definition II.3, the authors define the secrecy condition using difference of trace norm instead of trace distance. For easier comparison to other papers, it would be good if the authors could specify that the epsilon-security (or soundness) of QKD typically mentioned in other papers refer to $\epsilon = \epsilon^{\text{cor}} + 1/2 * \epsilon^{\text{sec}}$, to better highlight that trace norm difference is used here.

4. In Sec. IV.E, after equation IV.12, the authors introduced the state $|\text{ket}\{\nu^2\}$, but it does not appear to have been used in the manuscript. If it’s unnecessary, it would be good to remove.

5. In the supplementary notes, the complement of the events in some Ω for section A and section B uses different notations, $\neg\Omega$ (line before Eqn A.2) in section A and Ω^c in section B (first line of page 19). It would be good to standardise the notation used.

6. In Theorem C.2, the secrecy parameter is equivalent to $\max\{\epsilon_{PA} + 2\epsilon_s, \epsilon_a + 2\epsilon_{KV}\}$. This is different from the secrecy parameter in Theorem II.4, so this might be a typo.

7. In section F of the supplementary notes, the first equation and the line before that has probability given as q^x , with x in the superscript instead of the subscript everywhere else in the supplementary notes.

8. Minor language errors: In footnote 2, the second line has two “information”. In the line after Eqn IV.13, “entropy” is misspelled.

9. In page 4 (before the BB84 example), the authors claimed that the security proof for the “alternative protocol” where the parameter estimation step is done before the error correction, the security proof stays the same, except the reduction from Theorem II.4 to Claim IV.6 now follows almost trivially. If this is the case, would it be better to formulate the generic protocol using this “alternative protocol” structure since it also simplifies the security proof? Or is there any advantage in performing the error correction before the parameter estimation?

10. While the authors have clarified in Footnote 6 that the norm sign refers to the trace norm throughout the paper, I believe including the “subscript-1” on each norm sign would improve the readability of the paper.

11. When introducing the sample application of the technique on the B92 protocol, the authors claimed that the B92 protocol “has no natural entanglement-based analogue”. The authors should be more precise here since the two works that were cited in that paragraph, Refs [24, 33], analysed the security of B92 protocol by first converting it to an entanglement-based protocol.

12. The notation $\rho_{\{\wedge \Omega\}}$ is only properly introduced in Section IV.A while it has been used to define the secrecy criterion in the Results section. Perhaps the authors can explain the notation while defining the secrecy criterion.

I recommend that the manuscript be accepted for publication with minor revision pertaining to the issues I mentioned earlier. The simple-to-use tool provided in the manuscript would prove useful to readers who may be interested in analysing the security of QKD protocols, but find it difficult to understand the highly theoretical generalised EAT to analyse security against general attacks.

Reviewer #2 (Remarks to the Author):

COMMENTS FOR THE AUTHORS

In this work, the authors develop a security proof framework for quantum key distribution (QKD) capable of establishing general security bounds from i.i.d. security bounds systematically, as long as a certain sequentiality assumption is satisfied. The proposed technique, which is applicable to a broad class of protocols, is based on the generalized entropy accumulation theorem (GEAT). This makes it suitable for handling generic (device-dependent) prepare-and-measure protocols directly without switching to an entanglement-based version, in so surpassing a limitation of security proofs based on the original entropy accumulation theorem (EAT).

The paper is generally well-written, the results are clearly presented and the methodology is sound.

Major concerns

In order to better understand the contributions of the paper with respect to the existing literature, I would appreciate if the authors could clarify several concerns, which I list below.

(1) From the discussion in the right column of page 2, I understand the following. The original EAT requires to work in the entanglement-based (EB) setting, and therefore, its application to a PM protocol must necessarily rely on an EB formulation of the latter. However, this formulation requires to fix Alice's marginal state, and this is the main technical obstacle. On the contrary, the GEAT circumvents this problem by being compatible with the PM setting directly. Apparently, the limitation of the EAT with respect to PM protocols comes from its restricted model of side-information: the side information in the EAT is always freshly generated in every round, while the GEAT models the side information as a register held by Eve that is possibly updated from round to round. Both models seem general enough to deal with PM protocols, but in fact the former is not, and the reason is unclear to me. For instance, in the second paragraph of page 11 the authors claim that "*in a prepare-and-measure protocol, the adversary Eve intercepts the quantum state sent from Alice to Bob in each round and updates her side information based on that. Therefore, any technique used to deal with such protocols must allow for the side information to be updated like in the GEAT; the more restrictive scenario considered in the EAT does not capture this kind of protocol.*" But couldn't the $I_{\{j\}}$ of the original EAT formally model the updated side information? Where exactly is the loss of generality with regards to an actual protocol?

(2) The authors claim (page 5, right column): "*Applying the GEAT to entanglement-based protocols does not require Condition II.1. Hence, for protocols where Condition II.1 cannot be enforced, one can switch to an entanglement-based version instead and use our results for that setting presented in Supplementary Note C, although this will likely complicate the analysis.*" I guess, even when dealing with an EB protocol, EAT-type theorems require that future rounds leak no information about earlier rounds. Is this guaranteed if one removes the sequentiality condition II.1? In short, it would be useful for the reader to shortly discuss why using the EAT/GEAT with the EB version of a PM protocol allows to withdraw the sequentiality assumption and still consider fully general attacks (in this regard, the authors might consider adding a short discussion in Supplementary Note C, where the EB setting is specifically addressed).

(3) In fact, if I properly understand Supplementary Note C, one can reach the reduction of general attacks to collective attacks with an EB protocol as well, as long as the EB protocol sticks to the structure of Protocol 2. Does this not mean that:

(a) this reduction is possible within the original EAT,

(b) and more importantly, this reduction is possible without the sequentiality condition?

(4) Related to the former comment, the authors claim: "*it appears difficult or impossible to use the EAT to obtain reasonable finite-size key rates for all but the simplest prepare-and-measure protocols.*" This statement might be confusing or vague, since <https://arxiv.org/pdf/2203.06554.pdf> seems to be a counter-example to it. In that paper, the authors provide sufficient conditions on the public announcements of the protocol to guarantee the Markov conditions of the original EAT. It would be nice if the authors could clarify this.

(5) The authors explain how Condition II.1 (“*Eve can only be in possession of one of the systems $Q_{\{i\}}$ at the same time*”) can be practically met in several ways, one of them being that the signals be divided into blocks. If I understand properly, the underlying idea is that Eve cannot simultaneously interact with arbitrarily separated signals, such that putting together such distant signals in a common block naturally assures the validity of Condition II.1 inside every block. However, it would be nice if the authors could further elaborate on this proposal.

I find the assessment of the above issues crucial for a sensible recommendation to be made regarding the publication of the manuscript.

Minor concerns

Some extra minor concerns are presented next.

(1) In the last paragraph of page 3, the authors comment on the fact that, in the protocol, error correction goes first and parameter estimation follows (in the form of a statistical check). However, another comment is probably necessary, regarding the sifting. It would be nice to remark at this point that Alice’s RK function incorporates the sifting, if I understand it properly, by including a default symbol in its output alphabet to account for withdrawn rounds. For a reader only familiar with device-dependent protocols, it may be strange to see no reference to sifting in the text.

(2) In the first paragraph of page 4, the authors claim: “*The security proof stays exactly the same, except that the reduction from Theorem II.4 to Claim IV.6 now follows almost trivially and does not need the argument from Supplementary Note A.*” I find this comment a little bit disruptive with respect to the discussion, so I would suggest including it in a footnote instead.

(3) In the formulation of the BB84 protocol as an instance of Protocol 1, the authors introduce the notation $H^{\{x\}}$ for $x=0,1$. Formally, however, if the authors want to claim that $H^{\{0\}}=id$, this should be explicitly stated as a definition.

(4) In the last paragraph of the left column of page 5, the authors claim: “*we need to show that no matter how she tampers with the systems $Q_{\{i\}}$, either her tampering will be detected or else it does not allow her to gain information about Alice’s and Bob’s final key*”. This kind of statements seem a little bit vague for me, as they do not captivate the essence of QKD security: it is not just about detecting Eve, but about quantifying her maximum possible knowledge on the key. Of course, the latter must be essentially null if one refers to the final key ---as the authors do in the above statement--- but the claim is still not convincing for me. Is there anything technically wrong with replacing this claim by something like “we need to statistically upper bound Eve’s information gain about the raw key data irrespectively of her interaction with the quantum channel”? In my opinion, a statement in this line, not referred to the desired guarantee on the final key but to the role of the QKD parameter estimation step, is probably more convenient.

(5) At the beginning of the right column of page 10, some new notation is used. Namely, an “o” symbol is introduced (I believe, denoting the composition of maps), and an operation “ $\text{Tr}_{\{C_{\{i\}}\}}$ ” is presented (I believe, referring to the partial trace map). This notation may not be obvious for all readers, and it is good practice to explicitly present it in the text.

(6) In the right column of page 15, the authors claim: “*if the map EV from Protocol 1 has a particular structure that distinguishes between “test rounds”, in which Alice and Bob use their measurement outcomes to check whether Eve tampered with the protocol, and “data rounds”, in which Alice and Bob generate the raw data for their key, the derivation of a collective attack bound can be further simplified*”. I am not sure about what this statement means. Could the authors exemplify a QKD protocol that does not distinguish between test rounds and key rounds in the precise sense they are referring to in this comment?

Aesthetic concerns

Below I list some minor aesthetic modifications I suggest for the authors’ consideration:

- finite-sized → finite size (abstract)
- behave the same → behave identically and independently (abstract)
- Authentic classical channel → authenticated classical channel (first paragraph of the paper)
- prove its finite-size security → proving its finite-size security (last paragraph of page 1)
- information information → information (footnote 2)
- mutual information → mutual information (footnote 2)
- the actions Eve performs → the set of actions Eve performs (page 5, right column, first sentence)
- Typically, it is easy for Alice and Bob → In fact, it is easy for Alice and Bob (page 5, right column, I suggest to omit “typically” because I think the discussed feature, although simple, is actually not typical at all)
- Sample application → example of application (title of Appendix E)
- which is the case for → which is the case for (page 10, right column)

Reviewer #3 (Remarks to the Author):

The main result -- showing that security against collective attacks implies finite-size security against general attacks for a broad class of protocols -- is an important addition to the existing mathematical tools that are routinely used to assess security of QKD protocols. The result, to the best of my knowledge, is original. It builds upon the previous work of one of the authors (Renner). The paper is well written and the key points are clearly explained. I was not able to find any obvious flaws in the arguments. The methodology is appropriate and the maths seems to be correct. I recommend publication in the current form.

We thank the reviewers for their careful reading of our manuscript and their constructive comments. Below we respond in detail to all queries and comments.

Reviewer 1

In the manuscript submitted, the authors aims to provide a general method of analysing the security of quantum key distribution (QKD) against general attacks in the finite-size regime. They defined a general prepare-and-measure device-dependent QKD protocol framework, which encompasses many of the QKD protocols of interest to the community. The security of this general protocol framework is then analysed using the generalised entropy accumulation theorem (EAT), which reduces the analysis to security against a collective attack. Security against collective attacks is generally much easier to analyse, but for the unfamiliar, a numerical method to compute the collective attack bound (from earlier work) is provided. Their main result, presented in theorem II.4, provides a simple key rate formula of QKD secure against general attacks in the finite-size regime. As an application, they demonstrated the security of the B92 QKD protocol. Instructions on how to apply generalised EAT to entanglement based protocols and decoy state BB84 are also provided in the supplementary information.

The manuscript presents an important tool, in the form of theorem II.4, for researchers in the QKD community and beyond to perform security analysis of QKD protocols by simply fitting their protocols to the protocol framework and applying the key rate formula (Eqn II.2) along with the solving for collective attack bound (for instance using Eqn IV.14). This provides an avenue for the community, especially experimentalists and those working to implement the technology, to perform security analyses of their QKD protocols, without requiring deep understanding into the underlying generalised EAT, which is highly theoretical in nature. Interestingly, the tool provided is easier to utilise than previous results with EAT, where one must convert protocols into an entanglement-based protocol, which may not be easy to do, and applies to protocols with unbounded dimensional systems, which past technique such as the quantum de Finetti theorem cannot manage. One limitation of the proposed analysis, as mentioned in condition II.1, is the requirement that Eve can only possess one system at a time, which limits its applicability for certain applications. A workaround is briefly mentioned that can restore the analysis's validity by sacrificing the key rate.

Overall, the paper has no major issues. As far as I can tell, the claims are correct and the results presented here are interesting and useful for the community.

Response: We appreciate the reviewer's positive overall assessment of our work and are happy to respond to the specific points below.

1. In the first line of the main text, the authors mention "authentic classical channel". Do they mean an "authenticated classical channel"?

Response: Thanks for spotting this typo, we have corrected it.

2. In the first page (last paragraph of first column), the authors mentioned some general numerical techniques for analysing security against collective attacks. In addition to the work that the authors have mentioned, there are other works in this topic.

- (a) Wang et al, npj Quantum Information 5, 17 (2019)
- (b) Primateamaja et al, PRA 99, 062332 (2019)

- (c) Ref. [44] (Tan et al, npj Quantum Information 7, 158 (2021))
- (d) Brown et al, arXiv: 2106.13692 (2021)
- (e) Araujo et al, arXiv: 2211.05725 (2022)
- (f) Hu et al, Quantum 6, 792 (2022)

Response: We thank the reviewer for pointing us to these references, which are indeed relevant and which we now cite in our paper.

.....

3. In definition II.3, the authors define the secrecy condition using difference of trace norm instead of trace distance. For easier comparison to other papers, it would be good if the authors could specify that the epsilon-security (or soundness) of QKD typically mentioned in other papers refer to $\epsilon = \epsilon^{cor} + 1/2\epsilon^{sec}$, to better highlight that trace norm difference is used here.

Response: Thanks for the suggestion, we have added a sentence on this in the paragraph below the definition.

.....

4. In Sec. IV.E, after equation IV.12, the authors introduced the state $|\nu^2\rangle$, but it does not appear to have been used in the manuscript. If it's unnecessary, it would be good to remove.

Response: In this version of the proof $|\nu^2\rangle$ is indeed not used anymore, so we have removed it. Thank you for pointing this out.

.....

5. In the supplementary notes, the complement of the events in some Ω for section A and section B uses different notations, $-\Omega$ (line before Eqn A.2) in section A and Ω^c in section B (first line of page 19). It would be good to standardise the notation used.

Response: Thank you, we have fixed this and now use Ω^c throughout.

.....

6. In Theorem C.2, the secrecy parameter is equivalent to $max\epsilon_{PA} + 2\epsilon_s, \epsilon_a + 2\epsilon_{KV}$. This is different from the secrecy parameter in Theorem II.4, so this might be a typo.

Response: This is indeed just a typo, which we have now corrected.

.....

7. In section F of the supplementary notes, the first equation and the line before that has probability given as q^x , with x in the superscript instead of the subscript everywhere else in the supplementary notes.

Response: Thanks, we fixed this.

.....

8. Minor language errors: In footnote 2, the second line has two "information". In the line after Eqn IV.13, "entropy" is misspelled.

Response: Thanks, we fixed this.

.....

9. In page 4 (before the BB84 example), the authors claimed that the security proof for the “alternative protocol” where the parameter estimation step is done before the error correction, the security proof stays the same, except the reduction from Theorem II.4 to Claim IV.6 now follows almost trivially. If this is the case, would it be better to formulate the generic protocol using this “alternative protocol” structure since it also simplifies the security proof? Or is there any advantage in performing the error correction before the parameter estimation?

Response: Doing error correction before parameter estimation can improve the key rate, which can matter in practice. As a concrete example, consider BB84: if one does error correction first, Bob now knows his own key as well as the error-corrected key (which, by the raw key validation step, is equal to Alice’s key with high probability), so Bob can exactly count the number of bit-flips between his own and Alice’s key. On the other hand, if one does parameter estimation first, Alice and Bob would rely on a small subset of rounds to estimate the number of bit-flips, which yields a less precise estimate and can result in (slightly) worse key rates in the finite-size regime. Since doing error correction first is the more difficult case to prove, we decided to treat this case in the paper to show that this can be incorporated in our framework.

.....

10. While the authors have clarified in Footnote 6 that the norm sign refers to the trace norm throughout the paper, I believe including the “subscript-1” on each norm sign would improve the readability of the paper.

Response: Thanks for the suggestion, we agree that this might improve readability and have added the subscript to each norm.

.....

11. When introducing the sample application of the technique on the B92 protocol, the authors claimed that the B92 protocol “has no natural entanglement-based analogue”. The authors should be more precise here since the two works that were cited in that paragraph, Refs [24, 33], analysed the security of B92 protocol by first converting it to an entanglement-based protocol.

Response: We have clarified in the paper what we mean by “natural” (namely, that no artificial constraints on Alice’s reduced state are required) and explained that due to this, while Refs. [24,33] use an entanglement-based version, this results in significantly worse key rates.

.....

12. The notation $\rho_{\Lambda\Omega}$ is only properly introduced in Section IV.A while it has been used to define the secrecy criterion in the Results section. Perhaps the authors can explain the notation while defining the secrecy criterion.

Response: Thanks, we have added an explanation of the notation to Definition II.3.

.....

Comment # 13: I recommend that the manuscript be accepted for publication with minor revision pertaining the issues I mentioned earlier. The simple-to-use tool provided in the manuscript would prove useful to readers who may be interested in analysing the security of QKD protocols, but find it difficult to understand the highly theoretical generalised EAT to analyse security against general attacks.

Response: We again thank the reviewer for their helpful comments and hope that after our revisions, they will find the manuscript suitable for publication.

.....

Reviewer 2

In this work, the authors develop a security proof framework for quantum key distribution (QKD) capable of establishing general security bounds from i.i.d. security bounds systematically, as long as a certain sequentiality assumption is satisfied. The proposed technique, which is applicable to a broad class of protocols, is based on the generalized entropy accumulation theorem (GEAT). This makes it suitable for handling generic (device-dependent) prepare-and-measure protocols directly without switching to an entanglement-based version, in so surpassing a limitation of security proofs based on the original entropy accumulation theorem (EAT).

The paper is generally well-written, the results are clearly presented and the methodology is sound.

Response: We appreciate the reviewer’s overall assessment and are happy to respond to their specific comments below.

.....

Major concerns

(1) From the discussion in the right column of page 2, I understand the following. The original EAT requires to work in the entanglement-based (EB) setting, and therefore, its application to a PM protocol must necessarily rely on an EB formulation of the latter. However, this formulation requires to fix Alice’s marginal state, and this is the main technical obstacle. On the contrary, the GEAT circumvents this problem by being compatible with the PM setting directly. Apparently, the limitation of the EAT with respect to PM protocols comes from its restricted model of side- information: the side information in the EAT is always freshly generated in every round, while the GEAT models the side information as a register held by Eve that is possibly updated from round to round. Both models seem general enough to deal with PM protocols, but in fact the former is not, and the reason is unclear to me. For instance, in the second paragraph of page 11 the authors claim that “in a prepare-and-measure protocol, the adversary Eve intercepts the quantum state sent from Alice to Bob in each round and updates her side information based on that. Therefore, any technique used to deal with such protocols must allow for the side information to be updated like in the GEAT; the more restrictive scenario considered in the EAT does not capture this kind of protocol.” But couldn’t the I_i of the original EAT formally model the updated side information? Where exactly is the loss of generality with regards to an actual protocol?

Response: This is a good question and we have added some additional explanation on this to the paper. In short, the reason that the I_i in the original EAT cannot model Eve’s side information in a prepare-and-measure protocol is that these I_i -systems are produced and output in a round-by-round manner subject to the Markov condition of the EAT. This can be used for example to capture round-by-round information about basis choices published by Alice and Bob in an entanglement-based QKD protocol. However, in a prepare-and-measure protocol, Eve does not passively receive side information from every round, but can actively intervene in the protocol, change Alice’s message to Bob, and dynamically update her side information based on that (without necessarily outputting the side information in that particular round, as it may be used to decide on Eve’s attack in future rounds). This more “active” role of Eve in the protocol can unfortunately not be modeled with the I_i systems of the EAT. (As a sidenote, even modelling simple entanglement-based protocols with the EAT is slightly awkward (but possible), as evidenced by the somewhat unnatural assignment of systems in [1, Section 5.1].)

.....

(2) The authors claim (page 5, right column): “Applying the GEAT to entanglement-based protocols does not require Condition II.1. Hence, for protocols where Condition II.1 cannot be enforced, one can switch to an entanglement-based version instead and use our results for that setting presented in Supplementary Note C, although this will likely complicate the analysis”. I guess, even when dealing with an EB protocol, EAT-type theorems require that future rounds leak no information about earlier rounds. Is this guaranteed if one removes the sequentiality condition II.1? In short, it would be useful for the reader to shortly discuss why using the EAT/GEAT with the EB version of a PM protocol allows to withdraw the sequentiality assumption and still consider fully general attacks (in this regard, the authors might consider adding a short discussion in Supplementary Note C, where the EB setting is specifically addressed).

Response: There is an important difference between entanglement-based and prepare-and-measure protocols in terms of where Eve’s attack enters the model of the protocol: in entanglement-based protocols, Eve’s attack consists of preparing a certain input state, and the channels in the GEAT just model Alice’s and Bob’s actions (with some leakage of side information to Eve from the public communication between Alice and Bob). This means that the channels used in the GEAT are sequential irrespective of Eve’s attack, and the input state in the GEAT can be arbitrary, so there is no restriction on Eve’s attack. In contrast, in prepare-and-measure protocols Eve’s attack is part of the channels applied in the protocol, which necessarily need to have a sequential structure. This is where the sequentiality condition on Eve’s attack comes from. When converting from prepare-and-measure protocols to entanglement-based protocols, we really do give Eve more power by allowing her to “attack all rounds at once”, so the fact that there is no sequentiality condition on Eve’s attack in an entanglement-based protocol is not a technical quirk of the GEAT, but rather a genuine difference between these two models of attack. We agree that this is a good point to clarify and have added a paragraph to Supplementary Note C as suggested by the reviewer.

.....

(3) In fact, if I properly understand Supplementary Note C, one can reach the reduction of general attacks to collective attacks with an EB protocol as well, as long as the EB protocol sticks to the structure of Protocol 2. Does this not mean that:

- (a) this reduction is possible within the original EAT,
- (b) and more importantly, this reduction is possible without the sequentiality condition?

Response: Supplementary Note C indeed gives a generic reduction from general to collective attacks for entanglement-based protocols.

Regarding (a), it is not known whether this reduction is possible with the EAT in the same level of generality, but it might be. Unlike for prepare-and-measure protocols, where the EAT is clearly unsuitable due the active role that Eve plays in the protocol, for entanglement-based protocols it looks like the EAT is in principle applicable. The one caveat is that in the EAT, the side information needs to satisfy the Markov condition, which already makes applying the EAT to simple entanglement-based protocols rather awkward (see [1, Section 5.1], where even for the simple E91 protocol a non-trivial assignment of systems (top of p. 25 in [1]) is needed). This makes it unclear whether applying the EAT to generic entanglement-based protocols is possible, or whether additional structure is needed to satisfy the Markov condition. We note that [2] does provide a relatively generic reduction for entanglement-based protocols using the EAT subject to a condition on the public announcements that is necessary to satisfy the Markov condition (Sections 3.2.3 and 4.1 in [2]), but it is not clear whether these conditions can be weakened or removed without using the GEAT.

Regarding (b), indeed this reduction does not require the sequentiality condition, so in this respect it is stronger than the reduction for prepare-and-measure protocols. However, as we discuss in the paper, it is generally not possible to convert prepare-and-measure protocols to entanglement-based protocols without introducing artificial constraints in the latter that are hard

to combine with the GEAT (or EAT). Since most practical protocols are prepare-and-measure protocols, our direct proof for prepare-and-measure protocols is the only known generic technique for proving security of such protocols. This is why we consider the proof for prepare-and-measure protocols to be our main contribution and moved the discussion of entanglement-based protocols to the supplementary material.

.....

(4) Related to the former comment, the authors claim: “it appears difficult or impossible to use the EAT to obtain reasonable finite-size key rates for all but the simplest prepare-and-measure protocols”. This statement might be confusing or vague, since <https://arxiv.org/pdf/2203.06554.pdf> seems to be a counter-example to it. In that paper, the authors provide sufficient conditions on the public announcements of the protocol to guarantee the Markov conditions of the original EAT. It would be nice if the authors could clarify this.

Response: The important difference is that the paper mentioned by the reviewer only works for entanglement-based protocols (see [2, page 5] for a comment on the difficulty of applying their methods to prepare-and-measure protocols), whereas the comment in our paper talks about prepare-and-measure protocols. We have slightly rephrased the comment in our paper to make this clearer.

.....

(5) The authors explain how Condition II.1 (“Eve can only be in possession of one of the systems Q_i at the same time”) can be practically met in several ways, one of them being that the signals be divided into blocks. If I understand properly, the underlying idea is that Eve cannot simultaneously interact with arbitrarily separated signals, such that putting together such distant signals in a common block naturally assures the validity of Condition II.1 inside every block. However, it would be nice if the authors could further elaborate on this proposal.

Response: Yes, this is exactly the idea. We are currently working on this and other approaches to weakening the sequentiality requirement including a detailed analysis of finite-size rates. Implementing these approaches makes the protocol analysis more convoluted, so we would prefer to keep the submitted paper focused on the conceptually clean GEAT-based security proof and leave a detailed analysis of settings with a weakened sequentiality condition for future work.

.....

Minor concerns

(1) In the last paragraph of page 3, the authors comment on the fact that, in the protocol, error correction goes first and parameter estimation follows (in the form of a statistical check). However, another comment is probably necessary, regarding the sifting. It would be nice to remark at this point that Alice’s RK function incorporates the sifting, if I understand it properly, by including a default symbol in its output alphabet to account for withdrawn rounds. For a reader only familiar with device-dependent protocols, it may be strange to see no reference to sifting in the text.

Response: Thank you for this suggestion, we have added a few sentences on this in the discussion of the protocol.

.....

(2) In the first paragraph of page 4, the authors claim: “The security proof stays exactly the same, except that the reduction from Theorem II.4 to Claim IV.6 now follows almost

trivially and does not need the argument from Supplementary Note A.” I find this comment a little bit disruptive with respect to the discussion, so I would suggest including it in a footnote instead.

Response: Thanks, we agree and have moved this to a footnote.

.....

(3) In the formulation of the BB84 protocol as an instance of Protocol 1, the authors introduce the notation H^x for $x = 0, 1$. Formally, however, if the authors want to claim that $H^0 = id$, this should be explicitly stated as a definition.

Response: Thanks, we have added this for clarification.

.....

(4) In the last paragraph of the left column of page 5, the authors claim: “we need to show that no matter how she tampers with the systems Q_i , either her tampering will be detected or else it does not allow her to gain information about Alice’s and Bob’s final key”. This kind of statements seem a little bit vague for me, as they do not captivate the essence of QKD security: it is not just about detecting Eve, but about quantifying her maximum possible knowledge on the key. Of course, the latter must be essentially null if one refers to the final key—as the authors do in the above statement—but the claim is still not convincing for me. Is there anything technically wrong with replacing this claim by something like “we need to statistically upper bound Eve’s information gain about the raw key data irrespectively of her interaction with the quantum channel”? In my opinion, a statement in this line, not referred to the desired guarantee on the final key but to the role of the QKD parameter estimation step, is probably more convenient.

Response: Thank you for this suggestion, we have changed this to talking about bounding Eve’s information about the raw key as this might indeed be more intuitive.

.....

(5) At the beginning of the right column of page 10, some new notation is used. Namely, an “o” symbol is introduced (I believe, denoting the composition of maps), and an operation “ Tr_{C_i} ” is presented (I believe, referring to the partial trace map). This notation may not be obvious for all readers, and it is good practice to explicitly present it in the text.

Response: Thanks, we added this.

.....

(6) In the right column of page 15, the authors claim: “if the map EV from Protocol 1 has a particular structure that distinguishes between “test rounds”, in which Alice and Bob use their measurement outcomes to check whether Eve tampered with the protocol, and “data rounds”, in which Alice and Bob generate the raw data for their key, the derivation of a collective attack bound can be further simplified”. I am not sure about what this statement means. Could the authors exemplify a QKD protocol that does not distinguish between test rounds and key rounds in the precise sense they are referring to in this comment?

Response: Of course it is very common to distinguish between test and data rounds, which is why the technique in [3, Section V.A] has been developed especially for this setting and why we do this example explicitly in our paper. However, there are QKD protocols that do not have this

structure. For example, as we described in our answer to Comment (9) by Reviewer 1, one can construct a version of BB84 where the parameter estimation uses the error-corrected key instead of separate parameter estimation rounds, which can sometimes improve the key rate. In such a scenario there is no distinction between test and data rounds.

.....

Below I list some minor aesthetic modifications I suggest for the authors' consideration:

- finite-sized → finite size (abstract)
- behave the same → behave identically and independently (abstract)
- Authentic classical channel → authenticated classical channel (first paragraph of the paper)
- prove its finite-size security → proving its finite-size security (last paragraph of page 1)
- information information → information (footnote 2)
- mututal information → mutual information (footnote 2)
- the actions Eve performs → the set of actions Eve performs (page 5, right column, first sentence)
- Typically, it is easy for Alice and Bob → In fact, is easy for Alice and Bob (page 5, right column, I suggest to omit "typically" because I think the discussed feature, although simple, is actually not typical at all)
- Sample application → example of application (title of Appendix E)
- which is the for → which is the case for (page 10, right column)

Response: Thanks for these suggestions, we have implemented all of them in the paper (apart from the second-to-last one, where we personally prefer "Sample application").

.....

Reviewer 3

The main result – showing that security against collective attacks implies finite-size security against general attacks for a broad class of protocols – is an important addition to the existing mathematical tools that are routinely used to assess security of QKD protocols. The result, to the best of my knowledge, is original. It builds upon the previous work of one of the authors (Renner). The paper is well written and the key points are clearly explained. I was not able to find any obvious flaws in the arguments. The methodology is appropriate and the maths seems to be correct. I recommend publication in the current form.

Response: We are happy to hear that the reviewer appreciates the contributions in our paper and recommends publication.

.....

References

- [1] Frederic Dupuis, Omar Fawzi, and Renato Renner. Entropy accumulation. *Communications in Mathematical Physics*, 379(3):867–913, 2020.
- [2] Ian George, Jie Lin, Thomas van Himbeek, Kun Fang, and Norbert Lütkenhaus. Finite-key analysis of quantum key distribution with characterized devices using entropy accumulation. *arXiv preprint arXiv:2203.06554*, 2022.
- [3] Frédéric Dupuis and Omar Fawzi. Entropy accumulation with improved second-order term. *IEEE Transactions on information theory*, 65(11):7596–7612, 2019.

REVIEWER COMMENTS

Reviewer #1 (Remarks to the Author):

COMMENTS FOR THE AUTHORS

The authors have sufficiently addressed my concerns in the previous correspondence. However, Reviewer #2 has raised a valid concern regarding the sequentiality assumption that is necessary in security analyses via GEAT. This is in contrast to other techniques that do not rely on the sequentiality assumption (Condition II.1) which would then allow Eve to apply her attack simultaneously to all the signals exchanged in the protocol. As have been highlighted by Reviewer #2, sequential attack considered in GEAT is somewhat weaker than the so-called coherent attack (where Eve can coherently perform joint operation across multiple signals).

On the other hand, if Alice only sends the signal for the next round after Bob detects the signal for the present round, then sequential attack is indeed the most general attack that Eve can perform in this setting. However, this is not what is done in QKD experiments. In that case, the security proof presented in the current version of the manuscript cannot be directly applied to existing QKD experiments. This poses some limitations on the impact of the work.

That being said, in most cases, Eve will not have access to the signals from all n rounds (where n is the total number of rounds). Depending on the distance between Alice and Bob, the time taken for the signal to reach Bob's detector (in practice, this depends on the length of the channel and its refractive index) and the repetition rate of the source, it is possible to derive an upper bound, m , on the number of signals that Eve can access before Bob detects the signal. In some practical parameter regime, we will have $m < n$ and yet, the most general attack that Eve can perform is a coherent attack across these m rounds (instead of n rounds). As alluded by the authors in the manuscript, they can impose Condition II.1 on the level of blocks instead of rounds with some penalty. If the authors can provide more details on this approach (perhaps in the Supplementary Materials), I believe that this will give enough evidence to support the claim that we can apply GEAT to analyse the security of QKD against general attacks. Alternatively, as suggested by Reviewer #2, the authors can tone down their claim that the work provides a method to analyse the security of QKD against sequential attacks instead of general attacks.

REVIEWER COMMENTS

Reviewer #2 (Remarks to the Author):

COMMENTS FOR THE AUTHORS

I sincerely appreciate the authors' efforts to clarify all of my concerns. Their feedback was helpful for me to understand the contributions of the paper in detail.

According to the authors, the main contribution of this paper is to provide an EAT-based tool to address the security of prepare-&-measure (PM) protocols. This is indeed a novelty. In fact, compared to the existing tools, this approach might allow to address the security of device-dependent protocols without the need to invoke certain symmetries or dimension assumptions typically present in security analyses. At the same time, however, I deduce from the paper and the discussion with the authors that the suggested approach suffers from a couple significant drawbacks.

Although applicable to a wide range of protocols, the proposed technique relies on the sequentiality assumption, which is a strong one in PM protocols. Compared to the general setting of coherent attacks, this assumption leads to a notably restricted adversary model, unless one adopts drastic solutions that seem to totally or partially compromise the performance. Importantly, this sequentiality issue does not seem to be a superfluous loose end to be bypassed in the future. According to the authors, it lies at the heart of EAT-based analyses. On the contrary, existing tools often deployed in the device-dependent setup (such as the entropic uncertainty relation), may perhaps not be applicable to such a wide range of protocols, but do allow to address the coherent-attack setting in full generality. Because of this reason, I find it necessary to relax the main claim of the paper ---namely, that it provides a reduction of general attacks to collective attacks--- by replacing it with a more accurate one ---e.g., that it provides a reduction of sequential attacks to collective attacks---.

As another minor drawback, it also seems to me that the GEAT, in spite of being more suitable than the EAT for the assessment of device-dependent protocols, is still more cumbersome to use than most existing security analyses. This might limit its applicability.

In summary, despite the advantages that the GEAT may have with respect to the EAT, it is still unclear to me whether or not it is actually beneficial to analyze the security of PM protocols given the existing tools for that purpose. In my humble opinion, one gets the feeling that EAT-based analyses are simply more suitable for genuinely entanglement-based protocols, for which the sequentiality condition seems to be automatically satisfied.

Putting it all together, I do not think this work matches the scope of Nature Communications. Nevertheless, I believe the paper is suitable for publication in some form, as long as its main claim, which I discussed above, is conveniently relaxed to fit with its merits.

Reviewer 1

The authors have sufficiently addressed my concerns in the previous correspondence. However, Reviewer #2 has raised a valid concern regarding the sequentiality assumption that is necessary in security analyses via GEAT. This is in contrast to other techniques that do not rely on the sequentiality assumption (Condition II.1) which would then allow Eve to apply her attack simultaneously to all the signals exchanged in the protocol. As have been highlighted by Reviewer #2, sequential attack considered in GEAT is somewhat weaker than the so-called coherent attack (where Eve can coherently perform joint operation across multiple signals).

On the other hand, if Alice only sends the signal for the next round after Bob detects the signal for the present round, then sequential attack is indeed the most general attack that Eve can perform in this setting. However, this is not what is done in QKD experiments. In that case, the security proof presented in the current version of the manuscript cannot be directly applied to existing QKD experiments. This poses some limitations on the impact of the work.

That being said, in most cases, Eve will not have access to the signals from all n rounds (where n is the total number of rounds). Depending on the distance between Alice and Bob, the time taken for the signal to reach Bob's detector (in practice, this depends on the length of the channel and its refractive index) and the repetition rate of the source, it is possible to derive an upper bound, m , on the number of signals that Eve can access before Bob detects the signal. In some practical parameter regime, we will have $m < n$ and yet, the most general attack that Eve can perform is a coherent attack across these m rounds (instead of n rounds). As alluded by the authors in the manuscript, they can impose Condition II.1 on the level of blocks instead of rounds with some penalty. If the authors can provide more details on this approach (perhaps in the Supplementary Materials), I believe that this will give enough evidence to support the claim that we can apply GEAT to analyse the security of QKD against general attacks. Alternatively, as suggested by Reviewer #2, the authors can tone down their claim that the work provides a method to analyse the security of QKD against sequential attacks instead of general attacks.

Response: We thank the reviewer for reading our revised manuscript and are happy to hear that their concerns from the initial review have been addressed. We have provided a detailed discussion of the sequentiality condition in our response to Reviewer 2. As the reviewer points out, a natural and useful relaxation of the sequentiality condition is to allow Eve to be in possession of some number $s > 1$ of signals at a time. We briefly hinted at a way to incorporate this relaxed condition in the previous version of the paper by using a “block-based” analysis. We thank the reviewer for the encouragement to spell out this approach in more detail and have now added a new Supplementary Note C that provides the details.

.....

Reviewer 2

I sincerely appreciate the authors' efforts to clarify all of my concerns. Their feedback was helpful for me to understand the contributions of the paper in detail.

According to the authors, the main contribution of this paper is to provide an EAT-based tool to address the security of prepare-&-measure (PM) protocols. This is indeed a novelty. In fact, compared to the existing tools, this approach might allow to address the security of device-dependent protocols without the need to invoke certain symmetries or dimension assumptions typically present in security analyses. At the same time, however, I deduce from the paper and the discussion with the authors that the suggested approach suffers from a couple significant drawbacks.

Although applicable to a wide range of protocols, the proposed technique relies on the sequentiality assumption, which is a strong one in PM protocols. Compared to the general setting of coherent attacks, this assumption leads to a notably restricted adversary model, unless one adopts drastic solutions that seem to totally or partially compromise the performance. Importantly, this sequentiality issue does not seem to be a superfluous loose end to be bypassed in the future. According to the authors, it lies at the heart of EAT-based analyses. On the contrary, existing tools often deployed in the device-dependent setup (such as the entropic uncertainty relation), may perhaps not be applicable to such a wide range of protocols, but do allow to address the coherent-attack setting in full generality. Because of this reason, I find it necessary to relax the main claim of the paper —namely, that it provides a reduction of general attacks to collective attacks— by replacing it with a more accurate one —e.g., that it provides a reduction of sequential attacks to collective attacks—.

Response: We thank the reviewer for reading our revised manuscript and are happy to hear that their concerns from the initial review have been addressed. We agree that the sequentiality condition as stated in the previous version of our paper was rather strict and are grateful to the reviewer for allowing us to address this issue more comprehensively. Prompted by this, we have improved our results to only require a weakened version of the sequentiality assumption, which allows Eve to hold some number $s > 1$ of systems at a time, as also suggested by Reviewer 1.

More concretely, we have added a new Supplementary Note C that details how to relax the sequentiality assumption to allow Eve to hold some number $s > 1$ of signals at a time, instead of allowing only one signal as in the “strict” sequentiality assumption. This implements the “block analysis” that we had already hinted at in the previous version of the paper. This extension is a straightforward combination of the chain rule for min-entropies with our previous proof. As also pointed out by Reviewer 1, with this weakened sequentiality condition Alice and Bob can implement the protocol at whatever signal transmission frequency they like, and then simply pick s sufficiently large so that their implementation is covered by our security proof; picking a larger s does cause the second order terms to increase, but we obtain explicit and asymptotically optimal key rates for any constant s . This means that our work shows, for the first time, that a broad class of QKD protocols is secure against general attacks with asymptotically optimal performance, both in the sense of key bits generated per round of the protocol, as well as key bits generated per second. Given the generality of our results, we view this as an important contribution to the theory of quantum key distribution.

While we are confident that the weakened sequentiality condition will address the concrete concerns raised by the reviewer, we would also briefly like to expand on the reviewer’s comment that the sequentiality assumption “lies at the heart of EAT-based analyses”. Here, it is important to distinguish between the sequentiality of the channels in the GEAT (which is indeed inherent to (G)EAT-based analyses) and sequentiality conditions on the protocol. In our case, the GEAT channels are more or less identical to the channels implemented in the protocol (including Eve’s attack), which is why Eve’s attack “inherits” a sequentiality condition of the GEAT channels. However, GEAT-based security proofs do not necessarily have to use GEAT channels that are in such close correspondence with the actual protocol and Eve’s attack. The simplest example are entanglement-based protocols, which we treat in Supplementary Note D and which do not require a sequentiality condition on Eve’s attack (despite the GEAT channels of course still being sequential). However, it is also possible to envision ways of modelling prepare-and-measure protocols in a less direct way as GEAT channels, which would remove the need for sequential attacks entirely. Implementing this strategy for general protocols would require a significant strengthening of the GEAT, which might also be of independent information-theoretic interest.

.....

As another minor drawback, it also seems to me that the GEAT, in spite of being more suitable than the EAT for the assessment of device-dependent protocols, is still more cumbersome to use than most existing security analyses. This might limit its applicability.

Response: We agree that the GEAT is perhaps more cumbersome to use at first than e.g. the

very simple security proofs from entropic uncertainty relations, but on the other hand its generality means that once one is familiar with the technique, it can be applied easily to a broad range of protocols and one does not need to learn or invent new techniques for each new class of protocols.

.....

In summary, despite the advantages that the GEAT may have with respect to the EAT, it is still unclear to me whether or not it is actually beneficial to analyze the security of PM protocols given the existing tools for that purpose. In my humble opinion, one gets the feeling that EAT-based analyses are simply more suitable for genuinely entanglement-based protocols, for which the sequentiality condition seems to be automatically satisfied.

Putting it all together, I do not think this work matches the scope of Nature Communications. Nevertheless, I believe the paper is suitable for publication in some form, as long as its main claim, which I discussed above, is conveniently relaxed to fit with its merits.

Response: We hope that the updated version of our paper, and in particular Supplementary Note C on the weakened sequentiality condition, will convince the reviewer of the benefits of our approach.

.....

REVIEWER COMMENTS

Reviewer #1 (Remarks to the Author):

I think the author has sufficiently addressed the main concern regarding the limitation to sequential attacks. By allowing Eve to access s quantum systems at once, the authors have effectively prove security against general attacks. Reviewer 2 also raised a slight concern that the GEAT is more cumbersome to use compared to existing techniques and therefore, limit its applicability. I think this assessment is subjective and I personally think that someone who is familiar with the original EAT can readily perform security proofs with GEAT as most of the concepts are similar. Furthermore, EAT-like analyses may be unavoidable in some protocols (e.g., semi DI protocols, where the measurement device is untrusted, hence the Hilbert space dimension is unbounded).

Given its merits, I recommend publication of the work.

I also found a typo in Section C of the Supplementary Notes. In the paragraph after Condition C.1, we have "..., then send system Q_1 to Bob and receive Q_{s+1} from Bob" (the second Bob should be Alice)

Reviewer #2 (Remarks to the Author):

COMMENTS FOR THE AUTHORS

I sincerely acknowledge the efforts made by the authors to clarify my concerns and improve their manuscript. Indeed, the possibility of systematically lifting collective-attack bounds to general-attack bounds for device-dependent protocols might be a very remarkable theoretical advance. I still think some comments are in order though.

The authors suggest a “scheduling solution” for Alice and Bob to assure the validity of the sequentiality condition. If I am understanding it properly, the scheduling the authors suggest seems to attempt to impose, say, relativistic constraints on the arrival time of signals that might reveal if Eve’s intervention violates the sequentiality condition. I have concerns about the “scheduling alternative” for different reasons. For instance, (1) heuristically, what would be the necessary time resolution of the scheduling to hypothetically rule out Eve’s violation of the sequentiality condition (of course, without restricting her capability to speed up the signals)? Or, (2) how certain are we that speeding up the signals would be absolutely required for Eve to violate the assumption? It might be the case, but it is just not set in stone at the moment. What if, for instance, Eve decides that a certain signal only clicks if a later signal contains, say, 3 photons, or any other attack of this kind? This example does not require to speed up a signal, and still it might provoke that map $A_{\{i\}}$ does not stick to the required model $E'_{\{i-1\}}Q_{\{i\}} \rightarrow E'_{\{i\}}Q_{\{i\}}$ for your analysis to apply (i.e., the side-information does not seem to be updated in a sequential way). This is of course just an example.

In short, as long as multiple signals coexist in the channel, I think one must conservatively accept that sequentiality cannot be taken for granted or easily verified (please correct me if I am wrong). I believe this is a sensitive approach to follow. If my concerns are justified, you might prefer to shorten the discussion of the scheduling a little bit. It would probably be more cautious to simply say that enforcing the sequentiality condition could come at the price of compromising the performance, such that the block-wise analysis discussed in the new Supplementary Note C might be necessary in practice. Note that the fact that the authors do not discuss the block scenario in the main text is perfectly fine for me, in order to keep the presentation of the main contribution as simple as possible.

In addition to the suggestion above, I have one last comment regarding Supplementary Note C. It is nice that the asymptotic performance is not compromised by the block-type analysis. However, this (probably expected) feature is not totally conclusive in my opinion. To be precise, before I can recommend this significant piece of work for publication in Nature Communications ---which I am willing to do at the moment--- I find it necessary to complement Supplementary Note C with one figure exhibiting the finite-key+finite-block performance. If possible, a figure similar to Figure 1 in the main text illustrating a couple examples would be very nice, for some reasonable values of the block-size parameter and the total number of signals.

Reviewer 2

I sincerely acknowledge the efforts made by the authors to clarify my concerns and improve their manuscript. Indeed, the possibility of systematically lifting collective-attack bounds to general- attack bounds for device-dependent protocols might be a very remarkable theoretical advance. I still think some comments are in order though.

Response: We thank the reviewer for reading our revised manuscript and are happy to hear that we successfully clarified their concerns. We address their specific comments in detail below.

The authors suggest a “scheduling solution” for Alice and Bob to assure the validity of the sequentiality condition. If I am understanding it properly, the scheduling the authors suggest seems to attempt to impose, say, relativistic constraints on the arrival time of signals that might reveal if Eve’s intervention violates the sequentiality condition. I have concerns about the “scheduling alternative” for different reasons. For instance, (1) heuristically, what would be the necessary time resolution of the scheduling to hypothetically rule out Eve’s violation of the sequentiality condition (of course, without restricting her capability to speed up the signals)? Or, (2) how certain are we that speeding up the signals would be absolutely required for Eve to violate the assumption? It might be the case, but it is just not set in stone at the moment. What if, for instance, Eve decides that a certain signal only clicks if a later signal contains, say, 3 photons, or any other attack of this kind? This example does not require to speed up a signal, and still it might provoke that map A_i does not stick to the required model $E'_{i-1}Q_i \rightarrow E'_iQ_i$ for your analysis to apply (i.e., the side-information does not seem to be updated in a sequential way). This is of course just an example.

Response: Regarding point (1), extremely precise time controllers are already in use in commercial QKD setups (see e.g. here), so we do not expect this to be a problem in practice.

Regarding point (2), the observation that Eve can only violate the sequentiality condition by speeding up signals follows immediately from the causal structure of spacetime.¹ We have added a new figure (Figure 2) to the Supplemental Material to illustrate this and hope that this clarifies the question. Unfortunately, we were not able to understand the example mentioned by the reviewer, since to decide whether a certain signal clicks based on information in a later signal, Eve would have to send information about the later signal to Bob’s side before the earlier signal arrives, thus speeding up the signal. However, we hope that with our additional explanation, this is also resolved.

In short, as long as multiple signals coexist in the channel, I think one must conservatively accept that sequentiality cannot be taken for granted or easily verified (please correct me if I am wrong). I believe this is a sensitive approach to follow. If my concerns are justified, you might prefer to shorten the discussion of the scheduling a little bit. It would probably be more cautious to simply say that enforcing the sequentiality condition could come at the price of compromising the performance, such that the block-wise analysis discussed in the new Supplementary Note C might be necessary in practice. Note that the fact that the authors do not discuss the block scenario in the main text is perfectly fine for me, in order to keep the presentation of the main contribution as simple as possible.

¹As a minor semantic point, by “signal” here we mean anything that is correlated with the system sent out by Alice. For example, a possible attack Eve could run is the following: correlate (or entangle) a quantum system Y with a later signal just as it leaves Alice’s lab, then send the correlated system to Bob’s lab, and use it to run an attack on an earlier signal that is just about to enter Bob’s lab, violating the sequentiality condition. Here, we consider the quantum system Y as part of the later signal, so “speeding up the signal” is understood to include such scenarios. The relativistic constraints imposed by Alice’s and Bob’s schedule of course also rules out this kind of non-sequential attack.

Response: We hope that our response to the previous point clarifies that the (block-)sequentiality assumption can indeed be verified by Alice and Bob using a schedule for the signals.

.....

In addition to the suggestion above, I have one last comment regarding Supplementary Note C. It is nice that the asymptotic performance is not compromised by the block-type analysis. However, this (probably expected) feature is not totally conclusive in my opinion. To be precise, before I can recommend this significant piece of work for publication in Nature Communications —which I am willing to do at the moment— I find it necessary to complement Supplementary Note C with one figure exhibiting the finite-key+finite-block performance. If possible, a figure similar to Figure 1 in the main text illustrating a couple examples would be very nice, for some reasonable values of the block-size parameter and the total number of signals.

Response: Thank you for the suggestion to include key rate plots for the block scenario, too. For fibre-based implementations, determining the amount by which Eve can speed up the signal depends strongly on the specific fibre network, so this is hard to say in general (but can of course be calculated for any specific setup). However, for satellite-based QKD, the delay can be calculated in a principled manner (and is also much smaller than for fibre-based implementations, so that our technique, even with the block sequentiality condition, yields very good one-shot bounds). We have added a plot to the paper for a block size of 10, which is adequate for satellite-based QKD (as we explain in the paper), and another plot for a block size of 10^4 to illustrate what happens at larger block sizes.

Setting aside the finite-size analysis, we do not think that it was entirely expected that one can extend our analysis to the weakened sequentiality condition with any constant block size without a loss in asymptotic key rate, since no asymptotically tight reduction to i.i.d. at this level of generality was known prior to our work.

Finally, we are happy to hear that with this addition, the reviewer expects to be willing to recommend our manuscript for publication. We would again like to thank them for the constructive and detailed comments throughout the reviewing process, which have helped us improve our paper significantly.

.....

Reviewer #2 (Remarks to the Author):

COMMENTS FOR THE AUTHORS

I sincerely thank the authors for their great efforts and I address their comments one by one below.

A possible attack I had in mind goes as follows: consider a “late” pulse and an “early” pulse. Late pulse is entangled to a radio signal, this radio signal is sent through free-space and interacts with the “early” pulse. From the first response of the authors though, I see the authors consider this to be an instance of “speeding-up” signals. In fact, I agree that either scheduling or considering blocks may solve this problem.

As I said in my previous report, I am willing to recommend this work for publication. However, before I can do so, I would like the authors to address some final concerns, which I list below. I invite the authors to leave the manuscript untouched, unless my concerns trigger the need for some clarification.

- 1) I would like to better understand the no-signalling condition (its obscurity for me mainly comes from the fact that there does not seem to be an obvious equivalence between the physical systems and the systems that enter the GEAT). The question is as follows. Apparently, the condition demands that, for any given map M_i , the output state it yields for the side information system, E_i , can be reproduced by (1st) tracing out the system R_{i-1} and (2nd) applying a certain map from E_{i-1} to E_i . Does this mean that:
 - (i) the way the side information updates after any given round is independent of the quantum state sent through the channel in that roundOr only that...
 - (ii) the way the side information updates after any given round is only dependent on the quantum state sent through the channel in that round via the influence this state has on Eve’s probe in that round (which can be incorporated in the maps demanded by the no-signalling condition)
- 2) We only discussed the speeding-up of signals but...what about delaying signals, which is experimentally much simpler? Here’s my probably naive argument, which I would like the authors to invalidate, if possible. Let’s imagine that Eve delays signal 1 and makes it interfere with signal 2. Then, in principle, side-information in round 2 may carry information about signal 1 (namely, about Alice’s bit value $A_1!$), which is supposedly forbidden for EAT-type results. Why do attacks like these not compromise the no-signalling condition or the Markov condition, if it seems that new side-information is revealing information about earlier outputs?
- 3) The discussion on the scheduling in the right column of page 5 is still a little bit unclear to me. There are two unclear claims. The first one is: “*Alice and Bob can agree on a schedule on which signals are sent, and assuming that Eve cannot significantly speed up the transmission of signals, this would ensure that Condition II.1 is satisfied*”. The second one is “... (e.g. in satellite-to-satellite QKD), they travel at the speed of light and cannot be sped up further by Eve, so Condition II.1 can be enforced by sending signals on a pre-agreed schedule without issues”. These claims suggest that scheduling is necessary in free-space QKD, or if one assumes that Eve cannot speed up the transmission of signals. However, I understand “scheduling” as “lowering the frequency to assure that a sped-up signal cannot interfere with the previous one”. Hence, if signals cannot be sped-up at all (as is the case in free-space QKD), or if Eve is unable to speed-up signals by hypothesis, no scheduling (i.e. no lowering of the frequency) is required, apparently contradicting the quoted claims. As a final comment, I consider “scheduling” and “making blocks” as different solutions: one may be interested in making blocks in order to avoid lowering the frequency, but it will have an impact by increasing the effect of finite statistics. This is in

fact what I understand from Supplementary Note C, which does not seem to match the discussion in the right column of page 5 in the main text.

Lastly, I would like to comment on the newly added figure, Figure 4 (Figure 2 is very nice to illustrate the problem we already discussed). From the figure I see that the block-analysis is satisfactory for, say, ground-to-satellite QKD. However, separately post-processing 10^4 raw keys might be prohibitive in practice, thus, seemingly making the proposed analysis unsuitable for fiber-based QKD. Hence, as a concluding remark, I sincerely think the work presented by the authors has improved a lot now, because it exhibits a higher transparency: a reader may understand the benefits and generality of applying the GEAT to device-dependent protocols, and at the same time, be aware of the limitations that this approach has if compared to standard tools as the entropic uncertainty relation.

This said, I reaffirm my intention to recommend this work for publication upon satisfactory assessment of my questions above.

Minor concerns

I think I have spotted a typo in page 23, where the speeding up is discussed. Specifically, the formula for the step size seems to be " $s = \Delta_t \cdot \text{frequency}$ ", but the present formula reads " $s = \Delta_t / \text{frequency}$ ".

Reviewer 2

I sincerely thank the authors for their great efforts and I address their comments one by one below.

A possible attack I had in mind goes as follows: consider a “late” pulse and an “early” pulse. Late pulse is entangled to a radio signal, this radio signal is sent through free-space and interacts with the “early” pulse. From the first response of the authors though, I see the authors consider this to be an instance of “speeding-up” signals. In fact, I agree that either scheduling or considering blocks may solve this problem.

As I said in my previous report, I am willing to recommend this work for publication. However, before I can do so, I would like the authors to address some final concerns, which I list below. I invite the authors to leave the manuscript untouched, unless my concerns trigger the need for some clarification.

Response: Thank you very much for again carefully considering our revised manuscript and providing additional comments. We are happy to hear that our explanations resolved your question on other possible attacks and that you are willing to recommend our work for publication. We will provide detailed answers to your comments below.

I would like to better understand the no-signalling condition (its obscurity for me mainly comes from the fact that there does not seem to be an obvious equivalence between the physical systems and the systems that enter the GEAT). The question is as follows. Apparently, the condition demands that, for any given map M_i , the output state it yields for the side information system, E_i , can be reproduced by (1st) tracing out the system R_{i-1} and (2nd) applying a certain map from E_{i-1} to E_i . Does this mean that:

(i) the way the side information updates after any given round is independent of the quantum state sent through the channel in that round

Or only that... (ii) the way the side information updates after any given round is only dependent on the quantum state sent through the channel in that round via the influence this state has on Eve’s probe in that round (which can be incorporated in the maps demanded by the no-signalling condition)

Response: It can indeed sometimes be a little subtle to correctly identify the systems in the GEAT with the physical systems in the protocol. To understand the no-signalling condition, it is important to note that the channel that reproduces E_i without receiving R_{i-1} as input only needs to produce the correct marginal state on E_i , whereas of course the utility of E_i for Eve usually stems from its correlations with the outputs A_i and internal memory R_i . This is why option (i) is not the correct way to think about the condition: Eve’s side information will in general depend on the state sent through the channel, but the side information marginal can still be easy to reproduce. (As a simple example, consider the maximally entangled state between signal and side information. Clearly, the side information is correlated with the signal, but its marginal is just the maximally mixed state, which is easy to reproduce.) Option (ii) is a good way of thinking about it, but still a little too restrictive in general: as we point out in footnote 11, in prepare-and-measure setups the no-signalling condition is trivially satisfied because there is no R_{i-1} -system. Perhaps the most intuitive way to understand the no-signalling condition more generally is in a device-independent setup, where R_i would model the internal memory of an untrusted device. Then, the no-signalling condition says that Eve’s marginal future side information cannot be influenced by past states of the device’s internal memory. This basically means that the device cannot just “signal” information to Eve in the protocol. We hope that this helps in understanding this condition.

We only discussed the speeding-up of signals but...what about delaying signals, which is experimentally much simpler? Here's my probably naive argument, which I would like the authors to invalidate, if possible. Let's imagine that Eve delays signal 1 and makes it interfere with signal 2. Then, in principle, side-information in round 2 may carry information about signal 1 (namely, about Alice's bit value $A_1!$), which is supposedly forbidden for EAT-type results. Why do attacks like these not compromise the no-signalling condition or the Markov condition, if it seems that new side-information is revealing information about earlier outputs?

Response: Such attacks are ruled out by sending signals on a schedule, by which we mean sending the signal at a pre-agreed time (see also our answer to the question below). Hence, Bob knows when to expect Alice's i -th signal, so if Eve were to delay the signal, Bob would notice this. More formally, if Bob didn't receive a signal when he expected one due to Eve's delay, he would just record this as a "lost photon" outcome.

.....

The discussion on the scheduling in the right column of page 5 is still a little bit unclear to me. There are two unclear claims. The first one is: "Alice and Bob can agree on a schedule on which signals are sent, and assuming that Eve cannot significantly speed up the transmission of signals, this would ensure that Condition II.1 is satisfied". The second one is "... (e.g. in satellite-to-satellite QKD), they travel at the speed of light and cannot be sped up further by Eve, so Condition II.1 can be enforced by sending signals on a pre-agreed schedule without issues". These claims suggest that scheduling is necessary in free-space QKD, or if one assumes that Eve cannot speed up the transmission of signals. However, I understand "scheduling" as "lowering the frequency to assure that a sped-up signal cannot interfere with the previous one". Hence, if signals cannot be sped-up at all (as is the case in free-space QKD), or if Eve is unable to speed-up signals by hypothesis, no scheduling (i.e. no lowering of the frequency) is required, apparently contradicting the quoted claims. As a final comment, I consider "scheduling" and "making blocks" as different solutions: one may be interested in making blocks in order to avoid lowering the frequency, but it will have an impact by increasing the effect of finite statistics. This is in fact what I understand from Supplementary Note C, which does not seem to match the discussion in the right column of page 5 in the main text.

Response: Thanks for asking this. We have now also clarified this point in the paper in the section on *Modelling Eve's attack*: By "scheduling" we do not mean lowering the signal frequency, but rather sending the signals at a pre-agreed time. This is done anyway in QKD implementations because Alice and Bob need to agree on which signal was the i -th one (and they cannot just count because signals may get lost). In cases where Eve can speed up the signal, scheduling alone is no longer sufficient to ensure the sequentiality condition, so one can either lower the signal frequency or use the block technique. We hope that this and the small change in the paper clarify the question.

.....

Lastly, I would like to comment on the newly added figure, Figure 4 (Figure 2 is very nice to illustrate the problem we already discussed). From the figure I see that the block-analysis is satisfactory for, say, ground-to-satellite QKD. However, separately post-processing 10^4 raw keys might be prohibitive in practice, thus, seemingly making the proposed analysis unsuitable for fiber-based QKD. Hence, as a concluding remark, I sincerely think the work presented by the authors has improved a lot now, because it exhibits a higher transparency: a reader may understand the benefits and generality of applying the GEAT to device-dependent protocols, and at the same time, be aware of the limitations that this approach has if compared to standard tools as the entropic uncertainty relation. This said, I reaffirm my intention to recommend this work for publication upon satisfactory assessment of my questions above.

Response: We agree that the paper has benefitted a lot from the reviewers' detailed comments, so thank you!

.....

I think I have spotted a typo in page 23, where the speeding up is discussed. Specifically, the formula for the step size seems to be “ $s = \Delta * \text{frequency}$ ”, but the present formula reads “ $s = \Delta / \text{frequency}$ ”.

Response: Yes, thanks for spotting this!

.....

REVIEWERS' COMMENTS

Reviewer #2 (Remarks to the Author):

In the first place, I would like to thank the authors one more time for their detailed response.

According to their reply, the no-signalling condition is of a technical nature, but not physically restrictive in the way I had thought about.

Also, the newly added clarification about the scheduling alternative is very convenient, and I fully understand the argument about delaying signals: if a "detection time frame" is found empty by Bob, the corresponding round is discarded. To see that the prohibition that future rounds leak information about earlier rounds is compatible with a signal being delayed and undetected, one can assume that, for lost signals, the A_i is fully known to Eve, such that later rounds cannot leak "extra" information about them (say, trivial bound on the "accumulated entropy" of that round).

At this stage, the authors have satisfactorily addressed my concerns to a reasonable degree, and I would like to congratulate them again on their significant efforts to do so. Without further ado, I recommend this work for publication in Nature Communications.